# Addressing Spectral Energy Imbalance in Time-Series Forecasting with Gini-Guided Progressive Frequency Extraction

## Abstract

Time-series forecasting has recently seen growing interest, with increasing attention to frequency-domain representations. Real-world time-series often exhibit spectral distributions in which some frequency components have disproportionately large amplitudes. Since larger amplitudes correspond to higher energy, these components dominate the total energy. Such an imbalance biases models toward high-energy frequency components, preventing them from learning low-energy components, thereby harming generalization. We propose GiPFE (Gini-guided Progressive Frequency Extraction), a model-agnostic framework that progressively extracts high-energy frequency components from time-series. This progressive extraction is crucial because even after the strongest components are removed, the remaining parts may still contain relatively strong frequencies that sustain the imbalance. GiPFE measures the degree of spectral imbalance in each channel using the Gini coefficient and dynamically adjusts the number of components extracted at each stage to achieve precise extraction. By gradually separating dominant high-energy patterns, GiPFE prevents a single predictor from being dominated by a few strong components, allowing auxiliary lightweight heads to capture simple high-energy patterns while the backbone focuses on the remaining complex low-energy structures. Experiments on five real-world datasets with multiple backbone models demonstrate that GiPFE consistently improves forecasting performance across diverse architectures and domains. Our code is available at https://anonymous.4open.science/r/gipfe-E114.

## 1 Introduction

Time-series analysis has gained increasing attention in recent years, with applications in various domains such as smart manufacturing (Farahani et al., 2023), healthcare (Kaushik et al., 2020), and traffic prediction (Ermagun & Levinson, 2018). Since the introduction of Transformer models that address long-range dependencies, the field has rapidly progressed, and a variety of advanced models have been developed (Zhou et al., 2021; Nie et al., 2023; Liu et al., 2024b). Building on this progress, recent studies have increasingly explored frequency-domain representations to better understand the structural characteristics of time-series.

Various strategies have been explored to leverage frequency information. Early approaches apply frequency analysis as a preprocessing step to remove noisy components using hand-crafted filters. However, since these methods were designed for specific datasets, they struggle to adapt when data characteristics differ (Kumar et al., 2019; Song et al., 2021). Later models have incorporated frequency information by transforming the input sequence into the frequency domain and directly processing the entire spectrum within the model architecture (Yi et al., 2023; Xu et al., 2024). However, their architecture-specific designs make it difficult to extend them to other backbones. In addition, existing frequency-based models typically process the spectrum as a whole, without distinguishing between strong and weak components, leading to the fundamental issue discussed in the following.

When real-world time-series are transformed into the frequency domain by applying a Fourier transform, some components exhibit amplitudes much larger than others. In terms of energy (a measure of the overall influence on the signal), these dominant components account for disproportionately

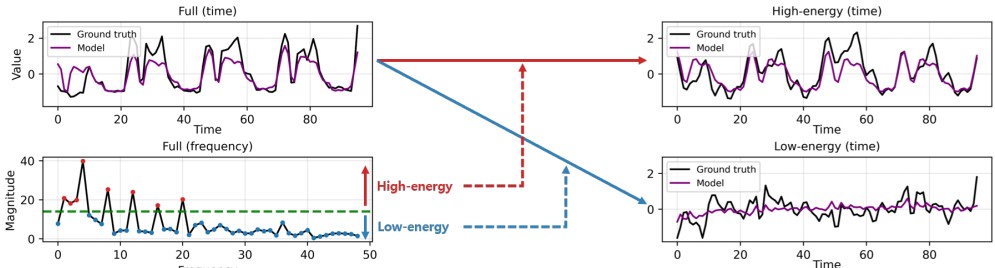

Figure 1: Example from the Electricity dataset on a single channel. Ground truth and model (DLinear) prediction are decomposed into high- and low-energy components based on the magnitude of the frequency spectrum. Low-energy components are less visible in the time domain and inadequately represented in the model predictions.

large portions of the total energy and contribute more strongly to the original signal. Consequently, high-energy components dominate the waveform, while low-energy components capture subtle variations. Because neural networks are sensitive to input magnitudes, high-energy components tend to dominate loss and gradient flow during training, whereas low-energy components are often underrepresented. As this imbalance persists throughout training, the model gradually fails to capture the full range of signal patterns, ultimately degrading generalization performance. For example, in the household electricity consumption scenario, high-energy patterns come from large appliances like air conditioners or refrigerators, while low-energy patterns arise from smaller devices such as lighting (U.S. Energy Information Administration, 2001). Though weak, these signals follow regular usage patterns, and ignoring them can lead to significant prediction errors. Figure 1 illustrates this issue with an example. In the time domain, the full signal (black) is dominated by the high-energy components (red), making the low-energy components (blue) difficult to recognize. Meanwhile, the model prediction (purple) partially follows the waveform of the high-energy components but fails to capture the variations of the low-energy components.

To address this limitation, we propose GiPFE (Gini-guided Progressive Frequency Extraction), which progressively extracts frequency components over multiple stages instead of processing the entire spectrum at once. At each stage, GiPFE estimates how unevenly the spectral energy is distributed using the Gini coefficient, and adjusts the number of components selected accordingly. This allows each stage to focus on components with similar energy scales, while passing weaker ones to subsequent stages. The extracted components are then processed according to their characteristics. The high-energy components extracted at each stage, which tend to exhibit simple and clear trends, are processed by lightweight MLP heads. In contrast, the residual gradually left behind across stages becomes increasingly composed of entangled low-energy components and is handled by the backbone, which can better capture their complex patterns. We also compute the spectral entropy of the high-energy and low-energy components for each dataset. The results show that high-energy components exhibit simpler structures, whereas low-energy components tend to be more complex. Detailed values are provided in Appendix C.

The contributions of this paper are as follows: 1) we propose GiPFE, which progressively extracts high-energy components to enable balanced, energy-aware processing of spectral components; 2) we design GiPFE as a model-agnostic module that can be seamlessly integrated into various time-series forecasting models in a plug-and-play manner, ensuring architectural generality and practical applicability; 3) we validate GiPFE on five real-world datasets using multiple different backbone models, demonstrating its effectiveness and broad applicability across diverse architectures and domains.

## 2 RELATED WORK

Time-series forecasting has evolved from classical statistical methods (Box et al., 2015) to recurrent neural network models (Hochreiter & Schmidhuber, 1997; Li et al., 2017; Yu et al., 2018; Salinas et al., 2020). More recently, Transformer architectures (Vaswani et al., 2017) have become the dominant paradigm for capturing long-range dependencies, with LogTrans (Li et al., 2019) and Informer (Zhou et al., 2021) demonstrating the scalability of attention mechanisms for long-sequence forecasting. DLinear (Zeng et al., 2023) and PatchTST (Nie et al., 2023) popularized channel-

independent modeling, achieving strong performance with remarkable efficiency and simplicity. Building on this trend, subsequent studies (Luo & Wang, 2024; Chen et al., 2024) introduced further architectural refinements. In parallel, iTransformer (Liu et al., 2024b) revisited channel-dependent designs, LLM-based approaches (Zhou et al., 2023; Jin et al., 2023; Liu et al., 2024a) transfer advances from natural language processing to time-series forecasting.

While most of these methods operate in the time domain, another line of research has focused on incorporating frequency-domain representations to enhance forecasting performance. Frequency-based approaches for time-series forecasting can be broadly categorized into two types: those that use frequency information as a preprocessing step and those that integrate it as part of the model. Preprocessing-based methods aim to suppress noise by removing certain frequency components from the input sequence, often relying on domain knowledge to predefine the filter bands. For example, Kumar et al. (2019) applied an FFT-based bandpass filter to emphasize QRS complexes in ECG signals, and Song et al. (2021) used Fourier transform denoising with thresholding to suppress market noise in financial time-series. However, such methods are difficult to generalize when the spectral energy distribution differs across datasets.

In contrast, recent approaches have incorporated frequency information directly into the model's internal mechanism. Several notable frequency-based models include FNet (Lee-Thorp et al., 2021), which replaces self-attention with a Fourier transform, FEDformer (Zhou et al., 2022a), which uses frequency-enhanced attention to capture long-range patterns, TimesNet (Wu et al., 2023), which leverages high-amplitude frequency components through CNN-based extraction, FiLM (Zhou et al., 2022b), which integrates frequency-domain projections to enhance memory modeling and suppress noise, FreTS (Yi et al., 2023), which transforms the input into the frequency domain and processes the entire spectrum with an MLP, and FITS (Xu et al., 2024), which performs interpolation over the full spectrum to generate future signals. Fredformer (Piao et al., 2024) converts the input into the frequency domain and applies patch-based tokenization along the spectral axis with a Transformer backbone. FilterNet (Yi et al., 2024), which transforms the input via FFT and applies learnable frequency filters to selectively amplify or attenuate specific spectral components. FreDF (Wang et al., 2025) transforms the label and prediction sequences into the frequency domain and introduces an auxiliary frequency-domain loss to correct the bias inherent in the Direct Forecast paradigm.

Although these methods do not rely on predefined frequency ranges and support end-to-end training, they process the spectrum as a whole, which fails to address the imbalance between high- and low-energy components. Some approaches can produce indirect effects on energy imbalance by alleviating the unevenness between low- and high-frequency components, but they do not explicitly resolve the imbalance from an energy perspective. Moreover, as complete forecasting models, they are tightly coupled to their specific architectures, making them difficult to transfer to other backbones.

## 3 PRELIMINARIES

**Problem formulation** We formulate the time-series forecasting task as learning a mapping from an input window $\mathbf{X}_{t-L+1:t} \in \mathbb{R}^{L \times C}$ to a target window $\mathbf{X}_{t+1:t+H} \in \mathbb{R}^{H \times C}$, where $L$ is the input length, $H$ is the prediction horizon and $C$ is the number of variables. For simplicity, we denote each instance by $\mathbf{x} = \mathbf{X}_{t-L+1:t} \in \mathbb{R}^{L \times C}$, and its corresponding prediction by $\mathbf{y} = \mathbf{X}_{t+1:t+H} \in \mathbb{R}^{H \times C}$.

**Gini coefficient** The Gini coefficient quantifies inequality based on the Lorenz curve, with the line of perfect equality serving as a reference for comparison. Figure 2 illustrates this concept by showing the Lorenz curve, the line of equality, and the area between them. Given a population of ordered values $a_{(1)} \leq \cdots \leq a_{(n)}$, we define the partial sums as follows:

$$S_0 = 0, \qquad S_k = \sum_{j=1}^{k} a_{(j)}, \quad k = 1, \ldots, n. \qquad (1)$$

The Lorenz curve is obtained by plotting the points $\left( \frac{k}{n}, \frac{S_k}{S_n} \right)$ for $k = 0, \ldots, n$ and connecting them. Denoting the area

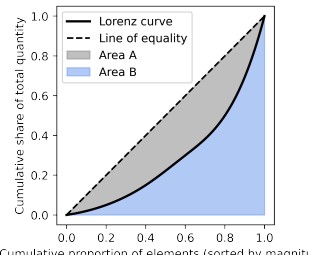

Figure 2: An example of the Lorenz curve and the line of equality.

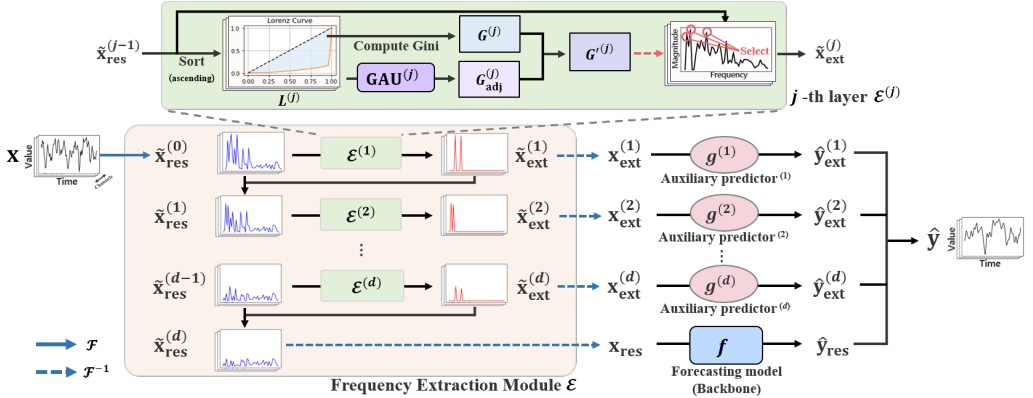

Figure 3: Overview of the proposed **GiPFE** framework. The blue arrows indicate the Fourier transform $\mathcal{F}$ and its inverse $\mathcal{F}^{-1}$, which convert between the time and frequency domains.

between the diagonal and the Lorenz curve as $A$, and the area under the Lorenz curve as $B$, the total area satisfies $A + B = \frac{1}{2}$. According to Sauerbrei (2009), the Gini coefficient is defined as

$$G = \frac{A}{A+B} = 2A = 1 - 2B = 1 - \frac{2}{n} \sum_{k=1}^{n} \frac{S_k}{S_n}. \tag{2}$$

## 4 METHOD

### 4.1 OVERVIEW

**GiPFE** (Gini-guided Progressive Frequency Extraction) is a model-agnostic framework that progressively extracts dominant frequency components while isolating the residual for forecasting. The extractor $\mathcal{E}$ operates through a depth of $d$ progressively stacked stages, where each stage isolates high-energy frequency components from the residual spectrum. Given an input sequence $\mathbf{x}$, GiPFE decomposes it into multiple extracted components and the remaining residual in a single call:

$$\{\mathbf{x}_{\text{ext}}^{(1)}, \mathbf{x}_{\text{ext}}^{(2)}, \ldots, \mathbf{x}_{\text{ext}}^{(d)}, \mathbf{x}_{\text{res}}\} = \mathcal{E}(\mathbf{x}). \tag{3}$$

Each extracted component $\mathbf{x}_{\text{ext}}^{(j)}$ is passed to a lightweight auxiliary predictor $g^{(j)}$, while the refined residual $\mathbf{x}_{\text{res}}$ is fed into the backbone model $f$:

$$\hat{\mathbf{y}}_{\text{ext}}^{(j)} = g^{(j)}(\mathbf{x}_{\text{ext}}^{(j)}), \qquad \hat{\mathbf{y}}_{\text{res}} = f(\mathbf{x}_{\text{res}}). \tag{4}$$

The final prediction is reconstructed as

$$\hat{\mathbf{y}} = \hat{\mathbf{y}}_{\text{res}} + \sum_{j=1}^{d} \hat{\mathbf{y}}_{\text{ext}}^{(j)}. \tag{5}$$

This design separates the roles of the two pathways: $g^{(j)}$ handles the extracted high-energy components, while $f$ is responsible for the progressively refined residual. It is worth noting that high-energy components tend to exhibit simple and clear trends and can be predicted by lightweight MLP heads $g^{(j)}$. On the other hand, the residual consists of entangled low-energy components that require a more expressive backbone $f$. Since the heads $g^{(j)}$ operate in parallel with $f$, they do not incur significant computational bottlenecks. Figure 3 illustrates the overall architecture of GiPFE.

### 4.2 FREQUENCY EXTRACTION MODULE

The frequency extraction module progressively extracts high-energy components from the spectrum of a time-series. As mentioned earlier, high-energy components can overshadow low-energy components and hinder their learning, making it necessary to distinguish between them. However, even

after removing the most dominant components once, relatively stronger components remain among the weaker ones, creating another imbalance. This means that instead of isolating a single group of high-energy components, the module must identify and separate multiple groups of high-energy components distributed across different energy scales.

Two simple approaches could be considered for this purpose: (i) sorting the components by magnitude and splitting them evenly, and (ii) dividing them using predefined thresholds. However, both approaches fail to adapt to the characteristics of individual sequences. The first approach risks grouping moderately strong components with a few extremely powerful ones and thus fails to handle data with a highly skewed energy distribution. In the second approach, the fixed thresholds cannot adapt to the varying energy ranges of different sequences, making the grouping ineffective.

Therefore, it is necessary to partition the components dynamically according to the actual distribution. Furthermore, instead of dividing the components into energy-scale groups all at once, we seek to progressively separate them. At each extraction stage, we measure the spectral energy imbalance using the Gini coefficient, which determines the appropriate number of high-energy components to extract. This design allows the model to learn components of different energy scales in a balanced manner without being dominated by the most energetic ones, while effectively adapting to the characteristics of individual sequences. As a result, it can uncover detailed patterns that were previously overshadowed, leading to more detailed and generalizable representations.

The internal procedure of the extractor $\mathcal{E}$ is as follows. The extractor first computes the initial spectrum $\tilde{\mathbf{x}}_{\text{res}}^{(0)} = \mathcal{F}(\mathbf{x})$ by applying a Fourier transform to the input sequence $\mathbf{x}$. It then iteratively extracts high-energy frequency components from the residual spectrum across $d$ stages. Specifically, at each stage $\mathcal{E}^{(j)}$, the extractor takes the residual spectrum $\tilde{\mathbf{x}}_{\text{res}}^{(j-1)} \in \mathbb{C}^{F \times C}$ as input and progressively isolates the high-energy frequency components by building a soft mask $W^{(j)}$ that indicates the proportion of energy to extract. Here, $F$ denotes the number of frequency bins and $C$ is the number of channels.

$$\tilde{\mathbf{x}}_{\text{ext}}^{(j)} = \mathcal{E}^{(j)}(\tilde{\mathbf{x}}_{\text{res}}^{(j-1)}) = W^{(j)} \odot \tilde{\mathbf{x}}_{\text{res}}^{(j-1)}, \qquad \tilde{\mathbf{x}}_{\text{res}}^{(j)} = (1 - W^{(j)}) \odot \tilde{\mathbf{x}}_{\text{res}}^{(j-1)}, \qquad j = 1, \ldots, d. \quad (6)$$

Finally, the extracted and residual spectra are transformed back to the time domain as:

$$\mathbf{x}_{\text{ext}}^{(j)} = \mathcal{F}^{-1}(\tilde{\mathbf{x}}_{\text{ext}}^{(j)}), \qquad \mathbf{x}_{\text{res}} = \mathcal{F}^{-1}(\tilde{\mathbf{x}}_{\text{res}}^{(d)}). \quad (7)$$

In the following, we present the construction of the soft mask $W^{(j)}$ in detail.

**DC handling** As the first step, we remove the DC (direct current) component, which represents the global mean and does not contain temporal variations. The DC bin ($k = 0$) is excluded from energy analysis. After removing it, we obtain the non-DC spectrum.

$$\tilde{\mathbf{x}}_{\text{res}\backslash 0}^{(j-1)} = \tilde{\mathbf{x}}_{\text{res}}^{(j-1)}[1 :] \in \mathbb{C}^{N_f \times C}, \quad (8)$$

where $N_f = F - 1$ is the number of non-DC bins. All subsequent processes are performed only on this non-DC part. During filter construction, a DC weight of $1$ is prepended to ensure that the DC bin is always routed to the extracted branch, making the residual DC zero after the first stage.

**Gini-based energy analysis** To identify high-energy components, we measure the degree of energy concentration across the non-DC frequencies, for which we compute the magnitude spectrum.

$$\mathbf{m}^{(j)} = |\tilde{\mathbf{x}}_{\text{res}\backslash 0}^{(j-1)}| \in \mathbb{R}^{N_f \times C}, \quad (9)$$

and sort it in ascending order $\mathbf{m}_{(1)}^{(j)} \leq \cdots \leq \mathbf{m}_{(N_f)}^{(j)}$ along the frequency axis. Let the cumulative sums of the sorted magnitudes be

$$S_0^{(j)} = 0, \qquad S_k^{(j)} = \sum_{i=1}^{k} \mathbf{m}_{(i)}^{(j)}, \quad k = 1, \ldots, N_f. \quad (10)$$

We then define the normalized cumulative energy, corresponding to the Lorenz curve, as:

$$L_k^{(j)} = \frac{S_k^{(j)}}{S_{N_f}^{(j)}} \in [0, 1]^C, \quad k = 1, \ldots, N_f. \quad (11)$$

The Gini coefficient is computed as follows:

$$G^{(j)} = 1 - \frac{2}{N_f} \sum_{k=1}^{N_f} \frac{S_k^{(j)}}{S_{N_f}^{(j)}} = 1 - \frac{2}{N_f} \sum_{k=1}^{N_f} L_k^{(j)} \ \in \ [0,1]^C. \tag{12}$$

**Gini Adjustment Unit**   To enhance the robustness of the raw Gini coefficient $G^{(j)}$, we introduce a Gini Adjustment Unit (GAU) that refines the estimation of spectral energy concentration. Each GAU takes the normalized cumulative energy values $\{L_k^{(j)}\}_{k=1}^{N_f}$ from the residual spectrum as input and produces a channel-wise adjustment term:

$$G_{\text{adj}}^{(j)} = \text{GAU}^{(j)}\big(\{L_k^{(j)}\}_{k=1}^{N_f}\big) \in \mathbb{R}^C. \tag{13}$$

Each GAU is implemented as a two-layer MLP with a final Tanh activation. We adopt Tanh to keep the adjustment small and stable while allowing both positive and negative corrections around zero, which prevents the refined ratio from becoming unstable. The adjusted Gini ratio is then computed as follows:

$$G'^{(j)} = G^{(j)} + G_{\text{adj}}^{(j)}. \tag{14}$$

We also clip $G'^{(j)}$ into $[0,1]$ to ensure its validity. During backpropagation, the straight-through estimator (STE) (Bengio et al., 2013) is applied to the clipping operation so that gradients can pass through its non-differentiable boundary. The STE replaces the zero gradient of clipping with an identity or piecewise approximation, which is a standard technique for handling quantized or discontinuous operations.

**Selection mask construction**   To progressively extract spectral components in a scale-aware manner, we construct a soft selection mask that isolates the high-energy frequencies at each stage while leaving the lower-energy components in the residual for subsequent refinement. We first compute the cumulative energy explained by the top-$k$ frequency bins by summing the magnitudes $\mathbf{m}^{(j)} \in \mathbb{R}^{N_f \times C}$ in descending order:

$$E_k^{(j)} = \sum_{i=1}^{k} \mathbf{m}_{(N_f - i + 1)}^{(j)}, \quad k = 1, \ldots, N_f. \tag{15}$$

The total spectral energy is given by $E_{\text{total}}^{(j)} := E_{N_f}^{(j)} \in \mathbb{R}^C$. We define the binary mask $\mathbf{b}_k^{(j)}$, which selects the top frequency bins whose cumulative energy does not exceed the channel-wise threshold, as follows:

$$\mathbf{b}_k^{(j)} = \mathbb{I}\big((G'^{(j)} \odot E_{\text{total}}^{(j)}) > E_k^{(j)}\big), \tag{16}$$

where $\mathbb{I}(\cdot)$ denotes the indicator function (1 if true, 0 otherwise), and $\odot$ indicates element-wise multiplication. However, such hard selection is non-differentiable and hinders gradient propagation. We therefore approximate this ideal binary mask with a differentiable soft frequency selection mask $\mathbf{w}_k^{(j)}$:

$$\mathbf{b}_k^{(j)} \approx \mathbf{w}_k^{(j)} = \sigma\big((G'^{(j)} \odot E_{\text{total}}^{(j)}) - E_k^{(j)}\big) \in \mathbb{R}^C, \quad k = 1, \ldots, N_f, \tag{17}$$

where $\mathbf{w}^{(j)} = [\mathbf{w}_k^{(j)}]_{k=1}^{N_f} \in \mathbb{R}^{N_f \times C}$. A smaller $G'^{(j)}$ (lower threshold) yields a mask that activates only a few strongest frequencies, while a larger $G'^{(j)}$ allows more frequencies to pass. As a result, bins in the top portion of the spectrum receive scores close to 1, while the lower-energy bins are gradually suppressed toward 0.

Finally, we prepend the DC bin to construct the full mask $W^{(j)}$. Since the DC bin ($k = 0$) represents the global mean component and is not involved in the Gini-based selection, we set its weight to 1 and concatenate it with the non-DC mask:

$$W^{(j)} = \text{concat}\big(1, \ \mathbf{w}^{(j)}\big) \in \mathbb{R}^{F \times C}. \tag{18}$$

This soft mask is used to extract high-energy frequency components at each stage (Eq. 6).

**Computational complexity** The frequency extraction module involves FFT, sorting, cumulative–energy computation, and mask construction. Among these, the FFT contributes the dominant cost of $O(CL \log L)$, while the remaining operations add only lower–order terms. Consequently, the module's overall complexity is $O(CL \log L)$. A detailed derivation is provided in Appendix D. Thus, the decomposition preserves the same asymptotic complexity as the FFT.

Moreover, transformer-based models commonly used in time-series forecasting have a self-attention complexity of $O(L^2 d_{\text{model}})$, where $d_{\text{model}}$ denotes the hidden feature dimension of the transformer block. In channel-independent designs, the same self-attention operation is repeated for each channel, resulting in $O(CL^2 d_{\text{model}})$. Our module adds an additional $O(CL \log L)$ cost under both settings. In the channel-dependent configuration, the backbone self-attention operates at $O(L^2 d_{\text{model}})$, so the overall complexity becomes $O(L^2 d_{\text{model}} + CL \log L)$, an asymptotic form consistent with transformer models that employ FFT-based preprocessing. In the channel-independent configuration, the backbone is dominated by $O(CL^2 d_{\text{model}})$, and the added $O(CL \log L)$ term does not change the asymptotic order of the backbone.

### 4.3 LOSS FUNCTION

The model is trained by supervising its predictions against the target sequence decomposed by the same extractor $\mathcal{E}$. We apply $\mathcal{E}$ to the ground truth target $\mathbf{y}$ once to obtain:

$$\{\mathbf{y}_{\text{ext}}^{(1)}, \mathbf{y}_{\text{ext}}^{(2)}, \dots, \mathbf{y}_{\text{ext}}^{(d)}, \mathbf{y}_{\text{res}}\} = \mathcal{E}(\mathbf{y}). \tag{19}$$

Letting $\hat{\mathbf{y}} = \hat{\mathbf{y}}_{\text{res}} + \sum_{j=1}^{d} \hat{\mathbf{y}}_{\text{ext}}^{(j)}$, the final loss is defined as the sum of the component and residual signal supervision terms:

$$\mathcal{L} = \sum_{j=1}^{d} \text{MSE}\big(\hat{\mathbf{y}}_{\text{ext}}^{(j)}, \mathbf{y}_{\text{ext}}^{(j)}\big) + \text{MSE}\big(\hat{\mathbf{y}}_{\text{res}}, \mathbf{y}_{\text{res}}\big). \tag{20}$$

## 5 EXPERIMENTS

### 5.1 EXPERIMENTAL SETTINGS

In this section, we describe the experimental settings, including the datasets and forecasting backbones used for the evaluation. The implementation details are provided in Appendix A.

**Datasets** We evaluate our model on five commonly used multivariate time-series datasets: ETT, Weather, ExchangeRate, Electricity, and Traffic. A summary of the datasets is provided in Appendix B. All datasets are split into training, validation, and test sets in a 7:2:1 ratio. We evaluate forecasting performance across varying prediction horizons $H \in \{96, 168, 336, 720\}$, with a fixed input sequence length of $L = 96$ for a fair and consistent comparison across different backbones.

**Forecasting backbones** We compare representative forecasting models. **SCINet** (Liu et al., 2022) recursively downsamples time-series into subsequences and applies convolution and interaction to capture temporal features at multiple resolutions. **DLinear** (Zeng et al., 2023) decomposes the time-series into trend and residual components, applying channel-independent linear projections for efficient forecasting. **PatchTST** (Nie et al., 2023) segments time-series into patches and applies channel-independent self-attention to efficiently model long-term dependencies. **FITS** (Xu et al., 2024) transforms time-series into frequency domain and performs interpolation over the entire spectrum to generate future signals. **iTransformer** (Liu et al., 2024b) employs an inverted Transformer design to better capture multivariate dependencies. **Fredformer**(Piao et al., 2024) transforms inputs into the frequency domain and applies patch-based tokenization along the spectral axis. **FilterNet** (Yi et al., 2024) transforms inputs via FFT and uses learnable spectral filters to modulate specific frequency components.

### 5.2 MAIN RESULTS

**Effect of GiPFE on backbone models** We evaluate the effectiveness of GiPFE as a plug-in method by applying it to five representative backbone models introduced above. Table 1 reports

Table 1: Performance comparison of backbone models with and without GiPFE.

| Methods | SCINet | | +GiPFE | | DLinear | | +GiPFE | | PatchTST | | +GiPFE | | FITS | | +GiPFE | | iTransformer | | +GiPFE | | Fredformer | | +GiPFE | | FilterNet | | +GiPFE | |
|---|---|---|---|---|---|---|---|---|---|---|---|---|---|---|---|---|---|---|---|---|---|---|---|---|---|---|---|---|
| Metrics | MSE | MAE | MSE | MAE | MSE | MAE | MSE | MAE | MSE | MAE | MSE | MAE | MSE | MAE | MSE | MAE | MSE | MAE | MSE | MAE | MSE | MAE | MSE | MAE | MSE | MAE | MSE | MAE |
| ETTh1 96 | 0.401 | 0.456 | 0.361 | 0.425 | 0.368 | 0.423 | 0.361 | 0.425 | 0.360 | 0.423 | 0.361 | 0.425 | 0.377 | 0.430 | 0.361 | 0.426 | 0.370 | 0.431 | 0.363 | 0.426 | 0.368 | 0.426 | 0.367 | 0.428 | 0.384 | 0.441 | 0.364 | 0.427 |
| ETTh1 168 | 0.475 | 0.509 | 0.392 | 0.451 | 0.398 | 0.449 | 0.390 | 0.450 | 0.395 | 0.452 | 0.394 | 0.452 | 0.413 | 0.459 | 0.390 | 0.450 | 0.399 | 0.456 | 0.386 | 0.447 | 0.403 | 0.455 | 0.391 | 0.451 | 0.419 | 0.469 | 0.391 | 0.452 |
| ETTh1 336 | 0.584 | 0.577 | 0.444 | 0.490 | 0.448 | 0.485 | 0.441 | 0.487 | 0.448 | 0.489 | 0.440 | 0.484 | 0.457 | 0.483 | 0.441 | 0.487 | 0.450 | 0.484 | 0.434 | 0.481 | 0.449 | 0.480 | 0.442 | 0.485 | 0.470 | 0.496 | 0.445 | 0.488 |
| ETTh1 720 | 0.682 | 0.635 | 0.621 | 0.602 | 0.556 | 0.560 | 0.607 | 0.594 | 0.582 | 0.578 | 0.575 | 0.577 | 0.593 | 0.564 | 0.608 | 0.594 | 0.600 | 0.571 | 0.577 | 0.578 | 0.615 | 0.580 | 0.577 | 0.578 | 0.623 | 0.581 | 0.582 | 0.580 |
| ETTh2 96 | 0.135 | 0.274 | 0.111 | 0.236 | 0.111 | 0.237 | 0.110 | 0.236 | 0.118 | 0.246 | 0.111 | 0.237 | 0.117 | 0.240 | 0.113 | 0.239 | 0.128 | 0.255 | 0.110 | 0.236 | 0.113 | 0.237 | 0.110 | 0.235 | 0.122 | 0.248 | 0.112 | 0.238 |
| ETTh2 168 | 0.171 | 0.308 | 0.128 | 0.254 | 0.126 | 0.253 | 0.128 | 0.253 | 0.134 | 0.262 | 0.126 | 0.251 | 0.136 | 0.255 | 0.128 | 0.253 | 0.140 | 0.264 | 0.126 | 0.252 | 0.132 | 0.253 | 0.127 | 0.251 | 0.138 | 0.261 | 0.126 | 0.250 |
| ETTh2 336 | 0.183 | 0.322 | 0.137 | 0.263 | 0.139 | 0.270 | 0.137 | 0.263 | 0.142 | 0.269 | 0.137 | 0.266 | 0.148 | 0.269 | 0.138 | 0.264 | 0.158 | 0.282 | 0.136 | 0.265 | 0.141 | 0.264 | 0.140 | 0.267 | 0.154 | 0.277 | 0.138 | 0.266 |
| ETTh2 720 | 0.219 | 0.354 | 0.163 | 0.290 | 0.175 | 0.313 | 0.163 | 0.288 | 0.168 | 0.299 | 0.162 | 0.290 | 0.186 | 0.292 | 0.162 | 0.287 | 0.196 | 0.308 | 0.161 | 0.290 | 0.180 | 0.295 | 0.160 | 0.289 | 0.196 | 0.308 | 0.163 | 0.295 |
| ETTm1 96 | 0.352 | 0.419 | 0.339 | 0.397 | 0.312 | 0.382 | 0.337 | 0.396 | 0.326 | 0.394 | 0.337 | 0.396 | 0.320 | 0.389 | 0.338 | 0.396 | 0.342 | 0.395 | 0.335 | 0.394 | 0.343 | 0.395 | 0.334 | 0.395 | 0.357 | 0.407 | 0.338 | 0.396 |
| ETTm1 168 | 0.397 | 0.448 | 0.366 | 0.420 | 0.356 | 0.410 | 0.368 | 0.420 | 0.356 | 0.413 | 0.363 | 0.418 | 0.366 | 0.417 | 0.368 | 0.420 | 0.375 | 0.422 | 0.361 | 0.416 | 0.366 | 0.418 | 0.363 | 0.417 | 0.382 | 0.427 | 0.364 | 0.418 |
| ETTm1 336 | 0.457 | 0.484 | 0.422 | 0.453 | 0.417 | 0.447 | 0.430 | 0.460 | 0.414 | 0.450 | 0.423 | 0.455 | 0.437 | 0.458 | 0.430 | 0.460 | 0.428 | 0.455 | 0.418 | 0.452 | 0.430 | 0.458 | 0.422 | 0.454 | 0.440 | 0.464 | 0.422 | 0.455 |
| ETTm1 720 | 0.562 | 0.553 | 0.477 | 0.495 | 0.471 | 0.489 | 0.475 | 0.492 | 0.465 | 0.489 | 0.471 | 0.490 | 0.499 | 0.500 | 0.474 | 0.491 | 0.500 | 0.505 | 0.474 | 0.491 | 0.499 | 0.506 | 0.471 | 0.490 | 0.507 | 0.509 | 0.473 | 0.492 |
| ETTm2 96 | 0.074 | 0.196 | 0.075 | 0.196 | 0.080 | 0.203 | 0.077 | 0.199 | 0.081 | 0.204 | 0.078 | 0.198 | 0.081 | 0.203 | 0.078 | 0.199 | 0.076 | 0.198 | 0.077 | 0.199 | 0.080 | 0.201 | 0.079 | 0.200 | 0.082 | 0.204 | 0.078 | 0.198 |
| ETTm2 168 | 0.103 | 0.238 | 0.092 | 0.217 | 0.093 | 0.219 | 0.092 | 0.218 | 0.097 | 0.218 | 0.092 | 0.218 | 0.097 | 0.221 | 0.092 | 0.219 | 0.093 | 0.221 | 0.092 | 0.219 | 0.096 | 0.221 | 0.093 | 0.218 | 0.097 | 0.223 | 0.092 | 0.218 |
| ETTm2 336 | 0.103 | 0.239 | 0.115 | 0.242 | 0.114 | 0.245 | 0.116 | 0.242 | 0.114 | 0.244 | 0.116 | 0.245 | 0.120 | 0.244 | 0.115 | 0.241 | 0.122 | 0.253 | 0.115 | 0.244 | 0.119 | 0.246 | 0.114 | 0.242 | 0.121 | 0.247 | 0.114 | 0.242 |
| ETTm2 720 | 0.153 | 0.294 | 0.140 | 0.266 | 0.143 | 0.275 | 0.140 | 0.266 | 0.142 | 0.267 | 0.139 | 0.268 | 0.151 | 0.269 | 0.140 | 0.266 | 0.152 | 0.275 | 0.139 | 0.268 | 0.151 | 0.271 | 0.139 | 0.268 | 0.156 | 0.276 | 0.139 | 0.268 |
| Weather 96 | 0.203 | 0.272 | 0.175 | 0.220 | 0.206 | 0.256 | 0.174 | 0.217 | 0.182 | 0.218 | 0.177 | 0.215 | 0.201 | 0.221 | 0.174 | 0.217 | 0.180 | 0.204 | 0.176 | 0.216 | 0.188 | 0.209 | 0.177 | 0.216 | 0.175 | 0.203 | 0.177 | 0.217 |
| Weather 168 | 0.246 | 0.303 | 0.211 | 0.258 | 0.245 | 0.301 | 0.209 | 0.254 | 0.215 | 0.254 | 0.210 | 0.246 | 0.235 | 0.247 | 0.210 | 0.256 | 0.218 | 0.234 | 0.214 | 0.255 | 0.226 | 0.239 | 0.214 | 0.256 | 0.212 | 0.232 | 0.215 | 0.257 |
| Weather 336 | 0.314 | 0.344 | 0.279 | 0.311 | 0.300 | 0.336 | 0.277 | 0.308 | 0.279 | 0.302 | 0.286 | 0.314 | 0.306 | 0.295 | 0.278 | 0.308 | 0.293 | 0.288 | 0.284 | 0.313 | 0.300 | 0.291 | 0.285 | 0.314 | 0.286 | 0.283 | 0.287 | 0.316 |
| Weather 720 | 0.367 | 0.370 | 0.345 | 0.357 | 0.359 | 0.382 | 0.340 | 0.347 | 0.345 | 0.358 | 0.347 | 0.350 | 0.364 | 0.332 | 0.341 | 0.348 | 0.358 | 0.328 | 0.346 | 0.350 | 0.364 | 0.332 | 0.347 | 0.350 | 0.351 | 0.324 | 0.348 | 0.352 |
| Exchange 96 | 0.088 | 0.219 | 0.055 | 0.170 | 0.052 | 0.164 | 0.055 | 0.171 | 0.054 | 0.168 | 0.055 | 0.172 | 0.053 | 0.165 | 0.055 | 0.170 | 0.056 | 0.169 | 0.055 | 0.171 | 0.053 | 0.164 | 0.054 | 0.170 | 0.056 | 0.172 | 0.055 | 0.172 |
| Exchange 168 | 0.128 | 0.263 | 0.090 | 0.217 | 0.089 | 0.217 | 0.087 | 0.214 | 0.185 | 0.341 | 0.089 | 0.219 | 0.086 | 0.213 | 0.090 | 0.217 | 0.090 | 0.217 | 0.090 | 0.219 | 0.086 | 0.213 | 0.090 | 0.220 | 0.092 | 0.223 | 0.090 | 0.220 |
| Exchange 336 | 0.205 | 0.338 | 0.174 | 0.306 | 0.178 | 0.312 | 0.164 | 0.297 | 0.350 | 0.476 | 0.166 | 0.301 | 0.175 | 0.310 | 0.164 | 0.297 | 0.186 | 0.320 | 0.166 | 0.302 | 0.180 | 0.315 | 0.166 | 0.302 | 0.186 | 0.320 | 0.166 | 0.302 |
| Exchange 720 | 0.419 | 0.490 | 0.294 | 0.406 | 0.362 | 0.461 | 0.260 | 0.382 | 0.604 | 0.633 | 0.296 | 0.414 | 0.498 | 0.498 | 0.261 | 0.383 | 0.387 | 0.484 | 0.300 | 0.417 | 0.385 | 0.481 | 0.300 | 0.416 | 0.481 | 0.554 | 0.299 | 0.416 |
| Electricity 96 | 0.186 | 0.299 | 0.180 | 0.265 | 0.195 | 0.277 | 0.180 | 0.265 | 0.196 | 0.278 | 0.181 | 0.265 | 0.203 | 0.280 | 0.180 | 0.265 | 0.190 | 0.269 | 0.181 | 0.265 | 0.191 | 0.271 | 0.185 | 0.271 | 0.181 | 0.272 | 0.181 | 0.265 |
| Electricity 168 | 0.204 | 0.312 | 0.175 | 0.265 | 0.183 | 0.272 | 0.176 | 0.265 | 0.184 | 0.271 | 0.176 | 0.264 | 0.187 | 0.272 | 0.176 | 0.266 | 0.184 | 0.268 | 0.179 | 0.269 | 0.184 | 0.270 | 0.177 | 0.265 | 0.181 | 0.273 | 0.176 | 0.265 |
| Electricity 336 | 0.223 | 0.334 | 0.188 | 0.283 | 0.197 | 0.294 | 0.189 | 0.283 | 0.197 | 0.293 | 0.189 | 0.284 | 0.202 | 0.290 | 0.189 | 0.284 | 0.199 | 0.287 | 0.190 | 0.285 | 0.207 | 0.300 | 0.190 | 0.286 | 0.199 | 0.295 | 0.191 | 0.286 |
| Electricity 720 | 0.252 | 0.355 | 0.221 | 0.320 | 0.233 | 0.332 | 0.222 | 0.320 | 0.228 | 0.326 | 0.222 | 0.320 | 0.239 | 0.324 | 0.221 | 0.320 | 0.240 | 0.325 | 0.222 | 0.321 | 0.251 | 0.339 | 0.223 | 0.321 | 0.244 | 0.337 | 0.224 | 0.323 |
| Traffic 96 | 0.494 | 0.429 | 0.401 | 0.331 | 0.505 | 0.388 | 0.400 | 0.331 | 0.434 | 0.354 | 0.410 | 0.338 | 0.521 | 0.386 | 0.400 | 0.331 | 0.397 | 0.332 | 0.410 | 0.338 | 0.384 | 0.318 | 0.382 | 0.325 | 0.447 | 0.359 | 0.412 | 0.340 |
| Traffic 168 | 0.446 | 0.382 | 0.412 | 0.332 | 0.479 | 0.366 | 0.412 | 0.332 | 0.450 | 0.356 | 0.420 | 0.338 | 0.495 | 0.366 | 0.412 | 0.332 | 0.427 | 0.342 | 0.421 | 0.338 | 0.418 | 0.340 | 0.421 | 0.339 | 0.452 | 0.357 | 0.428 | 0.344 |
| Traffic 336 | 0.441 | 0.372 | 0.437 | 0.346 | 0.504 | 0.381 | 0.437 | 0.346 | 0.472 | 0.366 | 0.446 | 0.352 | 0.525 | 0.381 | 0.437 | 0.346 | 0.451 | 0.355 | 0.446 | 0.352 | 0.445 | 0.352 | 0.446 | 0.352 | 0.478 | 0.372 | 0.451 | 0.356 |
| Traffic 720 | 0.495 | 0.399 | 0.470 | 0.370 | 0.533 | 0.408 | 0.469 | 0.370 | 0.495 | 0.386 | 0.477 | 0.374 | 0.552 | 0.402 | 0.469 | 0.370 | 0.482 | 0.380 | 0.477 | 0.374 | 0.488 | 0.388 | 0.476 | 0.374 | 0.514 | 0.397 | 0.484 | 0.380 |
| 1st Count | 2 | 2 | 30 | 31 | 0 | 10 | 24 | 23 | 9 | 11 | 23 | 23 | 4 | 9 | 28 | 23 | 3 | 10 | 30 | 1 | 4 | 13 | 28 | | 4 | 5 | 29 | 28 |
| Avg | 0.305 | 0.368 | 0.265 | 0.327 | 0.279 | 0.338 | 0.263 | 0.325 | 0.288 | 0.346 | 0.265 | 0.326 | 0.289 | 0.334 | 0.265 | 0.325 | 0.277 | 0.331 | 0.264 | 0.326 | 0.276 | 0.329 | 0.264 | 0.326 | 0.287 | 0.338 | 0.266 | 0.328 |

Table 2: Forecasting performance (MSE) of GiPFE compared to other model-agnostic methods (FAN, SAN, RevIN). GiPFE consistently achieves the best or competitive results, demonstrating its robustness and general applicability. Detailed results are provided in Appendix H.

| Models | DLinear | | | | | PatchTST | | | | | FITS | | | | | iTransformer | | | | |
|---|---|---|---|---|---|---|---|---|---|---|---|---|---|---|---|---|---|---|---|---|
| Methods | GiPFE | FAN | SAN | RevIN | No | GiPFE | FAN | SAN | RevIN | No | GiPFE | FAN | SAN | RevIN | No | GiPFE | FAN | SAN | RevIN | No |
| ETTh1 | 0.450 | 0.452 | 0.455 | 0.462 | 0.443 | 0.443 | 0.447 | 0.450 | 0.461 | 0.446 | 0.450 | 0.454 | 0.457 | 0.460 | 0.460 | 0.440 | 0.442 | 0.440 | 0.455 | 0.455 |
| Weather | 0.250 | 0.256 | 0.249 | 0.271 | 0.278 | 0.255 | 0.260 | 0.250 | 0.259 | 0.255 | 0.251 | 0.256 | 0.249 | 0.277 | 0.277 | 0.255 | 0.259 | 0.246 | 0.259 | 0.263 |
| Exchange | 0.142 | 0.150 | 0.166 | 0.191 | 0.170 | 0.152 | 0.153 | 0.180 | 0.180 | 0.298 | 0.142 | 0.149 | 0.166 | 0.180 | 0.180 | 0.153 | 0.153 | 0.168 | 0.184 | 0.180 |
| Electricity | 0.192 | 0.195 | 0.200 | 0.207 | 0.202 | 0.192 | 0.195 | 0.200 | 0.196 | 0.201 | 0.192 | 0.198 | 0.208 | 0.208 | 0.208 | 0.193 | 0.197 | 0.201 | 0.199 | 0.203 |
| Traffic | 0.430 | 0.431 | 0.460 | 0.519 | 0.505 | 0.438 | 0.439 | 0.459 | 0.469 | 0.463 | 0.430 | 0.431 | 0.475 | 0.523 | 0.523 | 0.438 | 0.440 | 0.442 | 0.437 | 0.440 |
| Avg | 0.293 | 0.297 | 0.306 | 0.330 | 0.320 | 0.296 | 0.299 | 0.308 | 0.313 | 0.333 | 0.293 | 0.298 | 0.311 | 0.330 | 0.330 | 0.296 | 0.298 | 0.299 | 0.307 | 0.308 |

the forecasting performance with and without GiPFE on five datasets and four prediction horizons. In terms of MSE, GiPFE consistently improves most backbones. On average, SCINet achieves the largest relative gain ($\approx$13.4%), followed by FITS ($\approx$7.47%), PatchTST ($\approx$6.73%), FilterNet ($\approx$6.44%), and iTransformer ($\approx$5.04%), while DLinear shows modest but non-negligible gains ($\approx$4.72%) and Fredformer also benefits from GiPFE ($\approx$4.04%). Notably, FITS, Fredformer, and FilterNet, all of which are frequency-based, still benefit from GiPFE, suggesting that GiPFE complements existing frequency-domain designs. Across datasets, the largest gains are observed on ExchangeRate ($\approx$16.0%), with additional improvements in all other datasets. These results demonstrate that GiPFE can enhance diverse backbones without architectural modifications, indicating its model-agnostic nature.

**Comparison with other model-agnostic methods** We further compare GiPFE with three representative model-agnostic add-on approaches that decompose the input for separate processing to improve time-series forecasting backbones. RevIN (Kim et al., 2021) restores normalized inputs to their original scale via an inverse mapping. SAN (Liu et al., 2023) predicts slice-specific statistics for better adaptability. FAN (Ye et al., 2024) discards fixed frequency components identified in the Fourier domain. Table 2 shows that GiPFE achieves superior performance compared to other model-agnostic methods, demonstrating its robustness across datasets and backbones. This highlights the superiority of GiPFE's *dynamic approach* to resolving spectral imbalance over the *static adjustment* applied by existing methods.

**Impact on inference time** Table 3 shows the inference time of PatchTST and iTransformer with SAN and GiPFE at depths $d = 1, 2, 3$. In GiPFE, each extraction stage introduces an additional lightweight predictor, so increasing $d$ adds more predictors and consequently increases the computation. SAN adds a single predictor to estimate the mean and variance, which makes its overhead comparable to that of GiPFE at $d = 1$, where only one additional stage is used. As $d$ increases

Table 3: Inference time (seconds) of PatchTST and iTransformer across five datasets, showing the base models and the latency impact of adding SAN or the proposed GiPFE module with different extraction depths ($d = 1, 2, 3$).

| Model | ETTh1 | Weather | ExchangeRate | Electricity | Traffic |
|---|---|---|---|---|---|
| PatchTST | 0.0047 | 0.0030 | 0.0060 | 0.0050 | 0.0040 |
| + SAN | 0.0080 | 0.0051 | 0.0071 | 0.0059 | 0.0069 |
| + GiPFE($d = 1$) | 0.0072 | 0.0060 | 0.0064 | 0.0055 | 0.0079 |
| + GiPFE($d = 2$) | 0.0090 | 0.0065 | 0.0087 | 0.0065 | 0.0088 |
| + GiPFE($d = 3$) | 0.0102 | 0.0130 | 0.0107 | 0.0104 | 0.0090 |
| iTransformer | 0.0050 | 0.0050 | 0.0040 | 0.0060 | 0.0089 |
| + SAN | 0.0070 | 0.0070 | 0.0070 | 0.0109 | 0.0092 |
| + GiPFE($d = 1$) | 0.0067 | 0.0070 | 0.0060 | 0.0070 | 0.0092 |
| + GiPFE($d = 2$) | 0.0081 | 0.0075 | 0.0078 | 0.0106 | 0.0160 |
| + GiPFE($d = 3$) | 0.0131 | 0.0104 | 0.0104 | 0.0134 | 0.0159 |

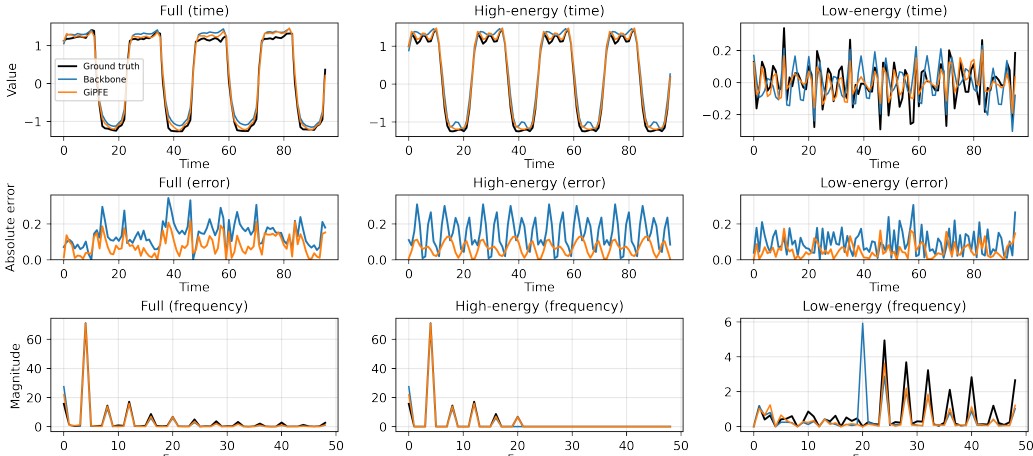

Figure 4: Example predictions on a single channel from the Electricity dataset, decomposed into high- and low-energy frequency components. We use PatchTST as the backbone and compare its predictions with those of the same backbone equipped with GiPFE.

to 2 and 3, the extra predictors in GiPFE accumulate and lead to larger increases relative to the single-stage variants. Even so, all configurations remain within the millisecond range, indicating that the added computation does not introduce meaningful latency and that GiPFE can be practically integrated into existing backbones.

**Visualization analysis**    We provide a visualization to examine whether our method can effectively isolate and predict the high- and low-energy components, and to evaluate its impact on enhancing the backbone predictions. In Figure 4, we present a detailed analysis of this effect by contrasting the ground truth and the model predictions after separating them into high- and low-energy components. In the low-energy part, the backbone fails to capture a substantial portion of low-energy components, while GiPFE compensates for the backbone by helping it capture some of the overlooked components. As a result, GiPFE shows a distribution closer to the ground truth than to the backbone. These results suggest that GiPFE corrects the predictions of the backbone, guiding it to learn a wider range of frequency components rather than overfitting to a few dominant ones, thereby enabling more stable and detailed predictions.

**Ablation study on frequency extraction strategy**    To verify the effectiveness of Gini-based adaptive extraction in GiPFE, we introduce two variants. The first, FixedSplit, simply divides the spectrum into three equally sized frequency bands (low-, medium-, and high-frequency) regardless of their energy distribution. The second, FixedThres, constructs two fixed thresholds at 2/3 and 1/3 of the maximum magnitude observed in the training dataset and decomposes the spectrum into three groups based on these thresholds. All other factors are kept the same as in GiPFE for a fair comparison. Unlike GiPFE, FixedSplit does not consider spectral imbalance and thus serves as a naive baseline for frequency-based feature extraction. In contrast, FixedThres relies on thresholds fixed from the training data, which prevents it from adapting to the spectral characteristics of individual

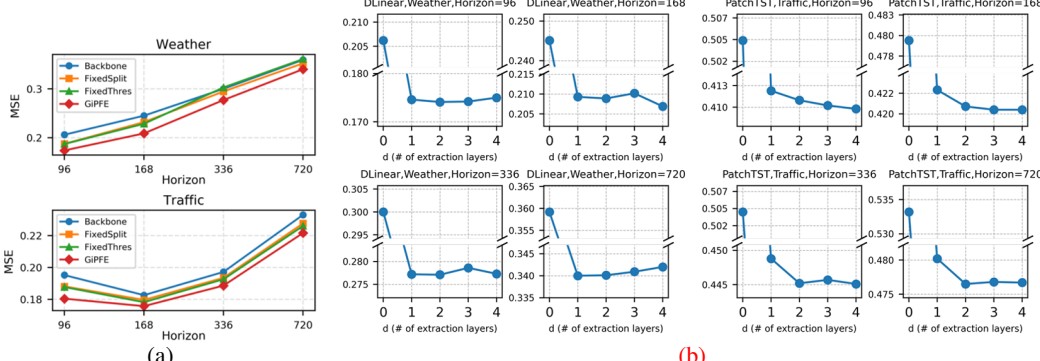

Figure 5: (a) Ablation study on the frequency extraction strategy. GiPFE consistently outperforms other variants on the Weather and Traffic datasets, evaluated using the DLinear backbone. (b) Sensitivity analysis on the number of extraction layers $d$ in GiPFE. The performance shows fluctuations as $d$ increases, suggesting the existence of an optimal range while confirming the robustness to the choice of $d$.

sequences. As shown in Figure 5 (a), while other variants provide a slight improvement over the backbone, GiPFE achieves the best performance by explicitly addressing spectral energy imbalance through its Gini-based adaptive extraction. This highlights that both resolving spectral imbalance and incorporating instance-wise adaptation to individual sequences are crucial in frequency-based feature extraction. Further analysis on the role of GAU, which complements these mechanisms, is provided in Appendix E, and an analysis of auxiliary predictor choices is presented in Appendix F.

**Sensitivity to the number of extraction layers** We analyze the sensitivity of GiPFE to the number of extraction layers $d$. Figure 5(b) reports the MSE trends for the Traffic dataset with the PatchTST backbone and for the Weather dataset with the DLinear backbone as $d$ varies. $d = 0$ corresponds to the plain backbone. Applying even a single extraction stage at $d = 1$ already yields a substantial improvement over $d = 0$, indicating that the proposed approach is effective even in the single-stage setting. However, the optimal value of $d$ varies across datasets and horizons because the degree of spectral energy imbalance differs among datasets. For datasets with more severe imbalance, a single stage may be insufficient to fully correct it; in such cases, increasing $d$ and applying the extraction progressively in a multi-stage manner helps mitigate the remaining imbalance and provides more stable performance gains. Once the performance enters a stable region, further increasing $d$ results in only negligible changes, meaning that beyond this point the model becomes largely insensitive to the choice of d and maintains consistent performance over a relatively wide range.

## 6 DISCUSSION

In this work, we propose GiPFE, a model-agnostic framework that progressively extracts high-energy frequency components to alleviate spectral imbalance and enhance forecasting generalization. Through extensive experiments, we demonstrated that both resolving spectral imbalance and incorporating instance-wise adaptation are crucial to fully exploiting frequency-based feature extraction, as evidenced by consistent improvements over strong baselines and ablation variants. At the same time, we note a minor practical consideration: as d increases, additional MLP layers are introduced at each extraction stage, which can modestly raise the overall computational cost. This effect is generally small but may be more noticeable in highly resource-constrained environments where only lightweight models such as DLinear are feasible. Meanwhile, pre-training settings with foundation models and online learning scenarios that require continual updates are emerging as key challenges. We consider extending the proposed approach to these settings as an important direction for future research.

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

Table 4: Statistics of the datasets.

| Datasets | ETTh1 | ETTh2 | ETTm1 | ETTm2 | Weather | Exchange | Electricity | Traffic |
|---|---|---|---|---|---|---|---|---|
| Variables | 7 | 7 | 7 | 7 | 21 | 8 | 321 | 862 |
| Length | 17,420 | 17,420 | 69,680 | 69,680 | 52,696 | 7,588 | 26,304 | 17,544 |
| Sampling Frequency | 1 Hour | 1 Hour | 15 Minutes | 15 Minute | 10 Minutes | 1 Day | 1 Hour | 1 Hour |

Ailing Zeng, Muxi Chen, Lei Zhang, and Qiang Xu. Are transformers effective for time series forecasting? In *Proceedings of the AAAI conference on artificial intelligence*, volume 37, pp. 11121–11128, 2023.

Haoyi Zhou, Shanghang Zhang, Jieqi Peng, Shuai Zhang, Jianxin Li, Hui Xiong, and Wancai Zhang. Informer: Beyond efficient transformer for long sequence time-series forecasting. In *Proceedings of the AAAI conference on artificial intelligence*, volume 35, pp. 11106–11115, 2021.

Tian Zhou, Ziqing Ma, Qingsong Wen, Xue Wang, Liang Sun, and Rong Jin. Fedformer: Frequency enhanced decomposed transformer for long-term series forecasting. In *International conference on machine learning*, pp. 27268–27286. PMLR, 2022a.

Tian Zhou, Ziqing Ma, xue wang, Qingsong Wen, Liang Sun, Tao Yao, Wotao Yin, and Rong Jin. FiLM: Frequency improved legendre memory model for long-term time series forecasting. In Alice H. Oh, Alekh Agarwal, Danielle Belgrave, and Kyunghyun Cho (eds.), *Advances in Neural Information Processing Systems*, 2022b.

Tian Zhou, Peisong Niu, Liang Sun, Rong Jin, et al. One fits all: Power general time series analysis by pretrained lm. *Advances in neural information processing systems*, 36:43322–43355, 2023.

# A  IMPLEMENTATION DETAILS

Each GAU is implemented as a two-layer MLP with hidden size 32 and a final Tanh activation. For performance, each lightweight auxiliary predictor $g^{(j)}$ takes not only the extracted component $\mathbf{x}_{\text{ext}}^{(j)}$ but also the original input $\mathbf{x}$. It first encodes $\mathbf{x}_{\text{ext}}^{(j)}$ through a small MLP with hidden dimension 64, concatenates the encoded features with $\mathbf{x}$, and then passes the fused representation through another MLP with hidden dimension 128 to produce the final prediction. We set $d$ to 3 for ETT, 2 for Weather, 3 for ExchangeRate, 2 for Electricity, and 3 for Traffic.

**Compute environment**   All experiments were carried out on a local machine running Windows 10, with an Intel Core i7-7700 CPU, 64GB RAM, and an NVIDIA Titan XP GPU.

# B  DATASET SUMMARY

Table 4 provides a summary of the basic statistics for each dataset used in our experiments. The **ETT**[1] dataset provides load and oil-temperature measurements collected from electricity transformers. It contains four standard subsets: **ETTh1**, **ETTh2**, **ETTm1**, and **ETTm2**. ETTh1 and ETTh2 offer hourly observations, whereas ETTm1 and ETTm2 provide the same variables sampled every fifteen minutes. **Weather**[2] contains 21 environmental variables sampled every 10 minutes. **ExchangeRate**[3] consists of daily currency exchange rates for eight countries. **Electricity**[4] comprises power usage data collected hourly from 321 users. **Traffic**[5] contains hourly readings from 862 freeway sensors in San Francisco.

---

[1] https://github.com/zhouhaoyi/ETDataset
[2] https://www.bgc-jena.mpg.de/wetter/
[3] https://github.com/laiguokun/multivariate-time-series-data
[4] https://archive.ics.uci.edu/ml/datasets/ElectricityLoadDiagrams20112014
[5] http://pems.dot.ca.gov

Table 5: Spectral entropy of the full spectrum, low-energy component, and high-energy component across datasets. The 2nd-level columns further decompose the low-energy component into its higher- and lower-energy portions.

| Dataset | Full | Low-energy | High-energy | Low-energy(2nd-level) | High-energy(2nd-level) |
|---|---|---|---|---|---|
| ETTh1 | 3.263 | 5.291 | 2.556 | 5.301 | 4.751 |
| Weather | 3.629 | 5.046 | 2.617 | 5.064 | 4.189 |
| ExchangeRate | 2.633 | 4.933 | 2.201 | 5.193 | 4.177 |
| Electricity | 3.011 | 5.412 | 2.655 | 5.510 | 5.105 |
| Traffic | 3.475 | 5.309 | 2.826 | 5.447 | 4.779 |

## C  SPECTRAL ENTROPY ANALYSIS OF HIGH- AND LOW-ENERGY COMPONENTS

This section presents the spectral entropy used to assess the complexity of the high-energy and low-energy components produced by our frequency partitioning scheme and describes how it is computed. For a time series segment $x \in \mathbb{R}^{L \times C}$ with length $L$ and $C$ channels, we estimate the power spectral density (PSD) of each channel $c$ using Welch's method:

$$\text{PSD}_c(f_k) = \text{Welch}(x_{:,c})[k]. \tag{21}$$

The PSDs are aggregated across channels to form the overall spectrum:

$$S(f_k) = \sum_{c=1}^{C} \text{PSD}_c(f_k). \tag{22}$$

We normalize this spectrum as

$$p_k = \frac{S(f_k)}{\sum_j S(f_j)}. \tag{23}$$

The spectral entropy is then expressed as a function of $x$:

$$H(x) = -\sum_k p_k \log p_k. \tag{24}$$

For each dataset, we report the average spectral entropy of the high-energy component and the low-energy component in Table 5. Across all datasets, the high-energy components consistently show lower entropy, indicating simpler spectral structures, whereas the low-energy components exhibit higher entropy, reflecting more complex and entangled patterns (Tenev et al., 2025). To further examine this trend, we additionally perform a second-level decomposition of the low-energy component. Although the absolute gap becomes smaller at this finer granularity, the same directional pattern remains: the subset with relatively higher energy consistently shows lower entropy than the subset with lower energy. This indicates that a residual form of energy–complexity imbalance persists even after the first-stage split, suggesting that a single-stage decomposition does not fully eliminate the imbalance. These observations provide supporting evidence for our progressive extraction strategy, as they show that meaningful imbalance remains in the residual components that are handled in later stages.

## D  COMPUTATIONAL COMPLEXITY OF THE FREQUENCY EXTRACTION MODULE

This appendix provides a detailed analysis of the computational cost of the proposed Frequency Extraction Module. The analysis follows the module's processing steps, including the initial spectrum

Table 6: Ablation on GAU (MSE). GiPFE* denotes GiPFE without GAU.

| Models | | DLinear | | |
|---|---|---|---|---|
| Dataset | Horizon | GiPFE | GiPFE* | Backbone |
| Weather | 96 | **0.1741** | 0.1754 | 0.2062 |
| | 168 | **0.2089** | 0.2125 | 0.2451 |
| | 336 | **0.2771** | 0.2793 | 0.3000 |
| | 720 | **0.3401** | 0.3432 | 0.3592 |
| Electricity | 96 | **0.1804** | 0.1809 | 0.1952 |
| | 168 | **0.1757** | 0.1770 | 0.1826 |
| | 336 | **0.1886** | 0.1895 | 0.1972 |
| | 720 | **0.2217** | 0.2226 | 0.2331 |

conversion, sorting-based energy analysis, Lorenz curve and Gini coefficient computation, GAU-based adjustment, adaptive mask generation, and final reconstruction.

The input sequence is first transformed into the frequency domain using an rFFT. Since the rFFT is applied to each channel with input length $L$, its computational cost is $O(CL \log L)$. The output frequency length is $F = L/2 + 1$.

After removing the DC component, the magnitude of each frequency–channel entry is computed. This requires element-wise operations over an $F \times C$ array and thus costs $O(CF)$. The magnitudes are then sorted for each channel, and sorting a length-$F$ vector per channel leads to a computational cost of $O(CF \log F)$.

Based on the sorted spectrum, the cumulative energy, the normalized energy progression (Lorenz curve), and the Gini coefficient are computed. Each of these operations is linear in the number of frequency elements, resulting in $O(CF)$ complexity. The GAU, which adjusts the Lorenz-derived information, processes a vector of length $F$ for each channel and therefore also costs $O(CF)$.

The adaptive soft mask is generated using the cumulative energy and the GAU output. Mask computation consists of element-wise operations and the restoration of the original frequency order, both of which remain within $O(CF)$. Finally, the masked spectrum and the updated residual spectrum are transformed back to the time domain using irFFT, which has the same computational cost as rFFT, namely $O(CL \log L)$.

Combining all operations yields

$$O(CL \log L) + O(CF \log F) + O(CF). \qquad (25)$$

Since $F = O(L)$, the final time complexity of the Frequency Extraction Module simplifies to

$$O(CL \log L). \qquad (26)$$

## E   ABLATION ON GAU

To isolate the effect of GAU, we compare *GiPFE* with *GiPFE\**, where GAU is removed while keeping all other components and training settings identical. As shown in Table 6, both GiPFE and GiPFE* substantially outperform the plain DLinear backbone across all horizons on Weather and Electricity, confirming that the Gini-based frequency extraction is the primary driver of improvement. The presence of GAU provides a further but relatively smaller gain, indicating that GAU serves as a complementary component that refines the extracted features rather than being the dominant factor.

## F   ABLATION ON AUXILIARY PREDICTOR

To examine the role of the auxiliary predictor, we compare our default lightweight MLP with a heavier PatchTST-based predictor while keeping all other components and training configurations

Table 7: Ablation on auxiliary predictor (MSE). $\text{GiPFE}^P$ uses PatchTST as the auxiliary predictor.

| Models | | PatchTST | | |
|---|---|---|---|---|
| Dataset | Horizon | GiPFE | $\text{GiPFE}^P$ | Backbone |
| Weather | 96 | **0.1767** | 0.1775 | 0.1816 |
| | 168 | **0.2099** | 0.2118 | 0.2154 |
| | 336 | 0.2856 | **0.2775** | 0.2789 |
| | 720 | 0.3467 | 0.3509 | **0.3447** |
| Electricity | 96 | **0.1809** | 0.1852 | 0.1955 |
| | 168 | **0.1757** | 0.1767 | 0.1838 |
| | 336 | 0.1889 | **0.1881** | 0.1969 |
| | 720 | 0.2217 | **0.2194** | 0.2284 |

Table 8: Forecasting performance (MSE) as mean ± standard deviation over 5 seeds.

| Models | | DLinear | | PatchTST | |
|---|---|---|---|---|---|
| Dataset | Horizon | +GiPFE | Backbone | +GiPFE | Backbone |
| Weather | 96 | **0.1744 ± 0.0007** | 0.2045 ± 0.0010 | **0.1792 ± 0.0016** | 0.1816 ± 0.0005 |
| | 168 | **0.2091 ± 0.0022** | 0.2430 ± 0.0048 | **0.2104 ± 0.0003** | 0.2150 ± 0.0010 |
| | 336 | **0.2777 ± 0.0007** | 0.3017 ± 0.0022 | 0.2861 ± 0.0010 | **0.2801 ± 0.0028** |
| | 720 | **0.3433 ± 0.0020** | 0.3585 ± 0.0004 | **0.3471 ± 0.0020** | 0.3478 ± 0.0056 |
| Electricity | 96 | **0.1805 ± 0.0001** | 0.1953 ± 0.0001 | **0.1811 ± 0.0001** | 0.1947 ± 0.0005 |
| | 168 | **0.1759 ± 0.0001** | 0.1825 ± 0.0001 | **0.1755 ± 0.0002** | 0.1841 ± 0.0003 |
| | 336 | **0.1887 ± 0.0002** | 0.1971 ± 0.0000 | **0.1891 ± 0.0004** | 0.1967 ± 0.0005 |
| | 720 | **0.2219 ± 0.0005** | 0.2331 ± 0.0002 | **0.2218 ± 0.0001** | 0.2283 ± 0.0003 |

unchanged. As shown in Table 7, the lightweight MLP already achieves stable performance across most horizons. As discussed in Appendix C, the high-energy components processed by the auxiliary predictor exhibit simple and periodic structures, meaning that more complex models offer limited additional benefit. Although the PatchTST auxiliary predictor yields modest improvements at longer horizons, the gains are small relative to the increased computational cost. For this reason, we adopt the lightweight MLP as the auxiliary predictor throughout the main paper.

## G ROBUSTNESS EVALUATION WITH MULTI-SEED EXPERIMENTS

To examine how sensitive the models are to changes in random seeds, we repeat experiments with five different seeds and report forecasting performance (MSE) using the mean and standard deviation. This provides a clearer view of the models' stability and variance characteristics. The corresponding results are presented in Table 8. As shown in the table, the proposed method maintains consistent variance levels across datasets and forecasting horizons, demonstrating robust performance.

## H DETAILED RESULTS FOR MODEL-AGNOSTIC COMPARISON

This section provides the detailed forecasting results (MSE) corresponding to Table 2. While Table 2 reports averages across datasets and horizons, Table 9 presents full-horizon results for each dataset. These results allow for a closer inspection of the performance of GiPFE compared with other model-agnostic methods (FAN, SAN, RevIN).

## I USE OF LARGE LANGUAGE MODELS

We used Large Language Models solely for grammar checking and polishing the writing.

Table 9: Forecasting performance (MSE) of GiPFE in detailed comparison with other model-agnostic methods (FAN, SAN, RevIN).

| Methods | | +GiPFE | +FAN | +SAN | +RevIN | DLinear | +GiPFE | +FAN | +SAN | +RevIN | PatchTST | +GiPFE | +FAN | +SAN | +RevIN | FITS | +GiPFE | +FAN | +SAN | +RevIN | iTransformer |
|---|---|---|---|---|---|---|---|---|---|---|---|---|---|---|---|---|---|---|---|---|---|
| ETTh1 | 96 | 0.3611 | 0.3661 | 0.3692 | 0.3759 | 0.3677 | 0.3611 | 0.3666 | 0.3663 | 0.3741 | 0.3604 | 0.3614 | 0.3657 | 0.3730 | 0.3772 | 0.3772 | 0.3632 | 0.3645 | 0.3619 | 0.3697 | 0.3697 |
| | 168 | 0.3901 | 0.3961 | 0.4060 | 0.4088 | 0.3983 | 0.3937 | 0.3989 | 0.4047 | 0.4089 | 0.3952 | 0.3900 | 0.3968 | 0.4120 | 0.4131 | 0.4131 | 0.3865 | 0.3940 | 0.3963 | 0.3991 | 0.3991 |
| | 336 | 0.4407 | 0.4420 | 0.4500 | 0.4567 | 0.4481 | 0.4404 | 0.4450 | 0.4528 | 0.4564 | 0.4485 | 0.4408 | 0.4432 | 0.4555 | 0.4568 | 0.4568 | 0.4337 | 0.4449 | 0.4304 | 0.4503 | 0.4503 |
| | 720 | 0.6070 | 0.6032 | 0.5935 | 0.6063 | 0.5559 | 0.5753 | 0.5780 | 0.5773 | 0.6049 | 0.5817 | 0.6075 | 0.6087 | 0.5869 | 0.5932 | 0.5932 | 0.5769 | 0.5650 | 0.5698 | 0.6001 | 0.6001 |
| Weather | 96 | 0.1741 | 0.1799 | 0.1732 | 0.1963 | 0.2062 | 0.1767 | 0.1831 | 0.1761 | 0.1767 | 0.1816 | 0.1743 | 0.1809 | 0.1732 | 0.2007 | 0.2007 | 0.1763 | 0.1830 | 0.1703 | 0.1776 | 0.1805 |
| | 168 | 0.2089 | 0.2159 | 0.2042 | 0.2297 | 0.2451 | 0.2099 | 0.2193 | 0.2063 | 0.2177 | 0.2154 | 0.2100 | 0.2136 | 0.2046 | 0.2355 | 0.2354 | 0.2138 | 0.2168 | 0.2027 | 0.2105 | 0.2183 |
| | 336 | 0.2771 | 0.2817 | 0.2786 | 0.3004 | 0.3000 | 0.2856 | 0.2901 | 0.2789 | 0.2909 | 0.2789 | 0.2780 | 0.2840 | 0.2788 | 0.3060 | 0.3059 | 0.2845 | 0.2875 | 0.2732 | 0.2919 | 0.2932 |
| | 720 | 0.3401 | 0.3457 | 0.3393 | 0.3586 | 0.3592 | 0.3467 | 0.3470 | 0.3378 | 0.3506 | 0.3447 | 0.3406 | 0.3452 | 0.3393 | 0.3644 | 0.3641 | 0.3462 | 0.3487 | 0.3382 | 0.3551 | 0.3582 |
| Exchange | 96 | 0.0554 | 0.0536 | 0.0549 | 0.0522 | 0.0520 | 0.0552 | 0.0540 | 0.0546 | 0.0521 | 0.0544 | 0.0546 | 0.0534 | 0.0550 | 0.0530 | 0.0530 | 0.0550 | 0.0549 | 0.0535 | 0.0552 | 0.0560 |
| | 168 | 0.0866 | 0.0872 | 0.0849 | 0.0882 | 0.0888 | 0.0889 | 0.0895 | 0.0880 | 0.0864 | 0.1853 | 0.0863 | 0.0874 | 0.0851 | 0.0867 | 0.0867 | 0.0896 | 0.0902 | 0.0877 | 0.0879 | 0.0901 |
| | 336 | 0.1644 | 0.1627 | 0.1715 | 0.1796 | 0.1784 | 0.1661 | 0.1663 | 0.1740 | 0.1744 | 0.3502 | 0.1643 | 0.1631 | 0.1717 | 0.1747 | 0.1748 | 0.1665 | 0.1669 | 0.1649 | 0.1938 | 0.1863 |
| | 720 | 0.2598 | 0.2952 | 0.3537 | 0.4454 | 0.3625 | 0.2964 | 0.3032 | 0.4016 | 0.4069 | 0.6040 | 0.2612 | 0.2926 | 0.3540 | 0.4068 | 0.4065 | 0.2998 | 0.3003 | 0.3656 | 0.4002 | 0.3869 |
| Electricity | 96 | 0.1804 | 0.1828 | 0.1886 | 0.1979 | 0.1952 | 0.1809 | 0.1835 | 0.1915 | 0.1801 | 0.1955 | 0.1803 | 0.1860 | 0.1979 | 0.2029 | 0.2028 | 0.1809 | 0.1838 | 0.1906 | 0.1879 | 0.1903 |
| | 168 | 0.1757 | 0.1788 | 0.1829 | 0.1837 | 0.1826 | 0.1757 | 0.1788 | 0.1833 | 0.1737 | 0.1838 | 0.1763 | 0.1822 | 0.1918 | 0.1875 | 0.1873 | 0.1790 | 0.1808 | 0.1832 | 0.1801 | 0.1841 |
| | 336 | 0.1886 | 0.1924 | 0.1980 | 0.2005 | 0.1972 | 0.1889 | 0.1928 | 0.1965 | 0.1906 | 0.1969 | 0.1888 | 0.1954 | 0.2052 | 0.2019 | 0.2017 | 0.1900 | 0.1952 | 0.1976 | 0.1937 | 0.1992 |
| | 720 | 0.2217 | 0.2278 | 0.2315 | 0.2449 | 0.2331 | 0.2217 | 0.2261 | 0.2289 | 0.2382 | 0.2284 | 0.2214 | 0.2298 | 0.2364 | 0.2395 | 0.2389 | 0.2225 | 0.2289 | 0.2314 | 0.2334 | 0.2398 |
| Traffic | 96 | 0.4001 | 0.4003 | 0.4422 | 0.5154 | 0.5049 | 0.4102 | 0.4115 | 0.4430 | 0.4399 | 0.4335 | 0.4000 | 0.4004 | 0.4516 | 0.5210 | 0.5207 | 0.4099 | 0.4113 | 0.4160 | 0.3940 | 0.3974 |
| | 168 | 0.4118 | 0.4117 | 0.4404 | 0.4905 | 0.4794 | 0.4205 | 0.4218 | 0.4381 | 0.4588 | 0.4504 | 0.4118 | 0.4118 | 0.4493 | 0.4955 | 0.4954 | 0.4208 | 0.4220 | 0.4265 | 0.4249 | 0.4268 |
| | 336 | 0.4371 | 0.4378 | 0.4635 | 0.5199 | 0.5045 | 0.4457 | 0.4454 | 0.4616 | 0.4803 | 0.4716 | 0.4373 | 0.4379 | 0.4744 | 0.5248 | 0.5247 | 0.4456 | 0.4457 | 0.4491 | 0.4503 | 0.4514 |
| | 720 | 0.4693 | 0.4723 | 0.4946 | 0.5485 | 0.5332 | 0.4768 | 0.4789 | 0.4926 | 0.4989 | 0.4954 | 0.4693 | 0.4724 | 0.5241 | 0.5524 | 0.5515 | 0.4767 | 0.4790 | 0.4745 | 0.4786 | 0.4825 |

