# OpenReview forum: "Addressing Spectral Energy Imbalance in Time-Series Forecasting with Gini-Guided Progressive Frequency Extraction"
_ICLR.cc/2026/Conference — Submitted to ICLR 2026_

### Official Review · Reviewer_gtry · 2025-10-29

**Soundness:** 2
**Presentation:** 2
**Contribution:** 2
**Rating:** 2
**Confidence:** 5

**Summary:**

The paper proposes GiPFE (Gini-guided Progressive Frequency Extraction), a model-agnostic add-on for time-series forecasting focusing on frequency modeling. It claims that spectral energy imbalance causes models to overfit high-energy frequency components. Experiments on five benchmarks demonstrated consistent gains and small inference-time overhead.

**Strengths:**

Frequency domain modeling is an important research topic for time series forecasting.

Leveraging a Gini-based progressive selection to adapt the number of extracted spectral components per instance is reasonable.

Paper is easy to follow.

**Weaknesses:**

**Motivation and paper positioning.**
The paper frames spectral energy imbalance and frequency modeling as the core research question, but several papers point out this exact direction like [1] Fredformer (KDD24), [2] FilterNet (NeurIPS 24), [3] FreDF (ICLR25). The Intro/Related Work sections emphasize general existing papers but do not properly situate against the above frequency modeling papers. Especially, I found the motivation in line 52-80 is largely overlap with Fredformer, a proper discussion and citation are needed. The paper itself claims most prior frequency models “process the spectrum as a whole,” but seems like several work explicitly departs from that. Overall, it is unclear what the actual new motivation of this paper is and what gap (or technical motivation) it is trying to fill beyond existing frequency modeling forecasting methods.

**Related work is insufficient.**
Most papers discussed in Related  Works section are in 2019–2023 with limited discussion of 2024–2025 frequency-oriented methods.

**Necessity of Gini.**
The method hinges on Gini coefficient as the imbalance metric (plus a learned GAU). The paper does not compare against other standard measures (e.g., spectral normalization or band-energy ratios), nor does it provide theory showing Gini is uniquely suitable for forecasting or optimization stability. Ablation is mainly w/ w/o GAU and fixed splits/thresholds. More comparisons across multiple imbalance metrics is needed.

**Model-agnostic framework.**
Architecturally, GiPFE is a pre/post decomposition with auxiliary heads and a residual path; this design is close in spirit to prior model-agnostic preprocessing/adaptation modules (e.g., RevIN/SAN/FAN). The paper compares with these three (Table 3/6), but the claim of being a general framework would be more convincing if (i) it robustly handled distribution shift (as RevIN/SAN aim to), (ii) it showed task-level generality beyond standard long-horizon forecasting, and (iii) it demonstrated compatibility with diverse frequency methods beyond simple addition to FITS. For me, GiPFE is better characterized as a plug-in feature extractor, not a full framework.

**Benchmarks and analyses are not yet sufficient.**
* ETT family: only ETTh1 is reported; standard practice evaluates ETTh1/ETTh2/ETTm1/ETTm2, or at least the four ETTh/ETTm variants for completeness and frequency diversity.
* variance/robustness: no std/error bars across runs; frequency-space methods are known to be sensitive to seeds and horizons.
* compute efficiency:  I think the improvements are margin compared to running time. Seems like general iTransformer can win the game in the Traffic dataset, but there is double inference time w/ GiPFE in Table 2.


**Minor** Several phrasing/style issues (e.g., “progres-” line breaks, small grammar nitpicks).

-------------------
- [1] Piao et al, Fredformer: Frequency Debiased Transformer for Time Series Forecasting, KDD 2024.
- [2] Yi et al, FilterNet: Harnessing Frequency Filters for Time Series Forecasting, NeurIPS, 2024.
- [3] Wang et al, FreDF: Learning to Forecast in the Frequency Domain, ICLR 2025.

**Questions:**

please refer to the above weakness.

---

> ### Author Response · Authors · 2025-11-20
> **Response to Reviewer gtry (1/2)**
>
> > **Motivation and paper positioning.** / **Related work is insufficient.**
>
> ### 1. Related work
>
> We thank the reviewer for pointing out these recent frequency-based works. They are not direct competitors and pursue different motivations, but they are relevant as related work, so we have added them with clearer descriptions. We also applied GiPFE on top of Fredformer and FilterNet, and Table 1 shows that our method provides complementary improvements.
>
> ### 2. Motivation
>
> As shown in the bottom-left panel of Fig. 1, our method focuses on a spectrum imbalance where high-energy and low-energy components become heavily skewed along the same frequency axis, represented conceptually by the horizontal green line separating stronger and weaker magnitudes. This is an energy-based imbalance defined by differences in magnitude above and below a horizontal threshold.
>
> In contrast, the high- vs. low-frequency issue in Fredformer corresponds to a different imbalance. It is defined by drawing an imagined vertical line on the spectrum and comparing the low-frequency region on the left with the high-frequency region on the right. Although they may seem related, the two target fundamentally different spectral properties.
>
> The consistent improvements observed when GiPFE is combined with Fredformer and FilterNet indicate that our method tackles a different problem. If the approaches handled the same type of imbalance, such gains would not appear. This shows that existing spectral models do not correct magnitude-based energy imbalance, and GiPFE complements them by addressing this missing component.
>
> > **Necessity of Gini.**
>
> To our knowledge, the cited examples (“spectral normalization’’ and “band-energy ratios’’) are not established metrics for measuring spectral energy imbalance. “Spectral normalization’’ refers to weight normalization in GANs, and “band-energy ratio’’ is not an imbalance metric in the time-series literature. From the context, it seems to describe the design directions of FilterNet and Fredformer rather than a formally defined measure. Our additional experiments show that GiPFE provides complementary gains to both FilterNet and Fredformer, confirming that Gini-based energy-imbalance mitigation addresses a distinct problem not handled by existing spectral models.
>
> Furthermore, the Gini coefficient is well suited to our problem because it directly measures inequality along the same frequency axis without requiring manually defined bands or thresholds. It provides a stable, scale-independent scalar that adapts to shifts in dominant components, giving a consistent criterion for masking across datasets. This makes it a natural choice for addressing energy imbalance and contributes to the stable improvements observed across backbones.
>
> Finally, if the reviewer had a specific “standard imbalance measure’’ in mind that differs from our interpretation above, we would greatly appreciate receiving its name or citation. We are more than willing to examine such a method and include additional comparisons.
>
> > **Model-agnostic framework.**
>
> The points raised in items (i)–(iii) extend beyond the scope of what our paper claims as its contributions, and they are not commonly considered necessary criteria for a framework in the long-term forecasting literature. For example, handling distribution shift is a distinct objective pursued by certain normalization-based methods, whereas GiPFE specifically targets spectral energy imbalance in the frequency domain. The fact that GiPFE does not explicitly address distribution shift reflects this difference in problem formulation rather than a methodological limitation.
>
> Similarly, task-level generality beyond forecasting is not a standard expectation in LTSF research, and existing model-agnostic modules referenced by the reviewer are also designed and evaluated solely for forecasting. It is therefore difficult to regard the absence of multi-task coverage as a weakness within this context.
>
> Finally, contrary to the reviewer’s concern about compatibility, we applied GiPFE not only to FITS but also to frequency-based models such as Fredformer and FilterNet, where it consistently improved performance. Moreover, GiPFE yielded similar gains when integrated with diverse non-frequency architectures including DLinear, PatchTST, and iTransformer. These results collectively demonstrate that GiPFE is broadly compatible with a wide range of backbones rather than being tied to a particular model family.

---

> ### Author Response · Authors · 2025-11-20
> **Response to Reviewer gtry (2/2)**
>
> > **Benchmarks and analyses are not yet sufficient.**
>
> We appreciate the reviewer’s comments regarding the completeness of our benchmarks and analyses. All three points have been addressed in the revised version as follows.
>
> ### 1. ETT family completeness
> The initial submission included only ETTh1. Following the reviewer’s suggestion, we expanded the evaluation to cover ETTh2, ETTm1, and ETTm2. The revised Table 1 now reports results on all four ETT variants, and GiPFE consistently improves performance across these settings.
>
> ### 2. Variance and robustness analysis
> We agree that frequency-space methods may be sensitive to seeds and horizons. To address this, we added 5-seed robustness experiments in Appendix G (Table 8), reporting mean and standard deviation. The results show that GiPFE maintains stable variance levels across datasets and horizons.
>
> ### 3. Compute efficiency
> Concerns about inference-time overhead are addressed in the revised Table 3. We report the absolute inference time of PatchTST and iTransformer with GiPFE at depths d=1,2,3 across five datasets. Although the relative increase exists, the absolute latency remains within the millisecond range, suggesting that the additional computation does not pose a practical burden.

---

### Official Review · Reviewer_t2LM · 2025-10-31

**Soundness:** 2
**Presentation:** 3
**Contribution:** 3
**Rating:** 4
**Confidence:** 2

**Summary:**

This paper introduces GiPFE, a model-agnostic framework designed to address spectral energy imbalance in time-series forecasting. The authors identify that real-world time-series often exhibit frequency components with disproportionately large amplitudes, causing models to overfit to high-energy components while neglecting low-energy patterns. GiPFE progressively extracts high-energy frequency components across multiple stages, using the Gini coefficient to measure spectral imbalance and dynamically adjust extraction. The extracted high-energy components are processed by lightweight MLP heads while the residual containing low-energy components is handled by the backbone model. Experiments on five datasets with multiple backbones demonstrate consistent improvements.

**Strengths:**

1 - The paper addresses a well-motivated and clearly articulated problem of spectral energy imbalance in time-series forecasting, providing concrete examples and visualizations that effectively illustrate how high-energy components can dominate learning and obscure low-energy patterns that contain important information.

2 - The proposed method is genuinely model-agnostic and demonstrates consistent improvements across diverse backbone architectures, with particularly impressive gains on some models, while maintaining minimal computational overhead.

3 - The use of the Gini coefficient for measuring spectral energy concentration is intuitive and well-justified, and the progressive extraction strategy with instance-wise adaptation is more sophisticated than fixed threshold or equal splitting approaches, as demonstrated in the ablation studies.

**Weaknesses:**

1 - The paper lacks theoretical analysis or justification for why progressive extraction should be superior to single-stage extraction, relying primarily on empirical results without providing theoretical insights into the optimization landscape or convergence properties of the proposed approach.

2 - The experimental setup appears limited with only 5 datasets tested, all from similar domains (electricity, weather, traffic), and no comparison with other frequency-based preprocessing methods beyond the three model-agnostic baselines, missing comparisons with methods like wavelet transforms or other spectral decomposition techniques.

3 - The hyperparameter selection process, particularly for the number of extraction stages d, seems dataset-specific and lacks a principled approach for determining optimal values, with the sensitivity analysis showing varying optimal values across horizons without clear guidance for practitioners.

4 - The paper does not discuss potential failure cases or limitations of the approach, such as scenarios where spectral imbalance might not be the primary challenge or datasets with different frequency characteristics like purely stochastic signals or white noise.

**Questions:**

1 - What is the computational complexity of the Gini coefficient calculation and soft mask construction at each stage, and how does this scale with the number of frequency bins and channels?

2 - Have you tested GiPFE on datasets with different frequency characteristics, such as financial time series with heavy stochastic components or audio signals?

3 - Could you provide theoretical analysis or bounds on why progressive extraction should outperform single-stage extraction?

---

> ### Author Response · Authors · 2025-11-20
> **Response to Reviewer t2LM**
>
> ## Weakness 1 & Question 3
>
> A general theoretical justification for the superiority of progressive extraction is not feasible due to the strong dataset-dependent variability in spectral energy and residual complexity. To address it empirically, we added a stepwise spectral entropy analysis in Appendix C. The results show that a single-stage split leaves a non-negligible imbalance and that the second-stage decomposition continues to reduce it. This provides empirical evidence for why progressive extraction can outperform a one-shot approach.
>
> ## Weakness 2 & Question 2
>
> Our dataset selection follows the standard LTSF benchmarking protocol. Recent work such as [1], [2], and [3] also adopts the same benchmark suite (ETT, Weather, Electricity, Traffic), indicating that this setup is well established rather than a weakness.
> ExchangeRate already covers the financial domain. To our knowledge, there is no established audio benchmark for long-term forecasting, but if such datasets become available, exploring this direction would be a meaningful extension for future work.
>
> Regarding wavelet-based baselines, our method requires fine-grained, bin-level frequency analysis. Wavelet transforms do not provide exact frequency coefficients but only scale-dependent bands, which makes them misaligned with the objective of our approach. For these reasons, we centered our comparison on FFT-based methods.
>
> Moreover, while several frequency preprocessing techniques are designed with specific architectures in mind, recent work also explores model-agnostic approaches. To provide a fair comparison, we include FITS, Fredformer, and FilterNet in Table 1 and show that GiPFE improves all of them, demonstrating compatibility with existing spectral models. If there is a specific additional baseline the reviewer has in mind, we are happy to include it.
>
> [1] Zeng, Ailing, et al. "Are transformers effective for time series forecasting?." AAAI 2023.
>
> [2] Nie, Yuqi, et al. “A Time Series Is Worth 64 Words: Long-Term Forecasting with Transformers.” ICLR 2023.
>
> [3] Huang, Songtao, et al. “TimeKAN: KAN-based Frequency Decomposition Learning Architecture for Long-Term Time Series Forecasting.” ICLR 2025.
>
> ## Weakness 3
>
> The extraction depth d is not a highly sensitive hyperparameter, as performance changes only mildly across a wide range of values. The optimal d differs across datasets because each dataset has a different degree of spectral energy imbalance, which determines how many extraction stages are beneficial. Only when d is set to an extremely small value do we observe noticeable degradation. Therefore, selecting d does not require extensive tuning and does not impose a practical burden on practitioners.
>
> ## Weakness 4
>
> First, to the best of our knowledge, long-term time-series forecasting tasks are generally designed to capture meaningful temporal structure rather than to predict purely stochastic patterns such as white noise. Since white noise contains no dependency between past and future, any model would naturally converge toward predicting its unconditional mean. Therefore, we view such cases as outside the scope of LTSF evaluation and not indicative of meaningful failure modes for our approach.
> In addition, beyond the reviewer’s point, we incorporated a clarification of the practical limitations of our method in the main text.
>
> ## Question 1
>
> The additional operations introduced by our module are all lower-order terms relative to the FFT cost $O(C L \log L)$, so the overall complexity remains fully characterized by $O(C L \log L)$. To make this clearer, we added a more explicit derivation and a consolidated explanation in the Computational complexity paragraph of Section 4.2 and in Appendix D.

---

> ### Comment · Reviewer_t2LM · 2025-11-25
>
> I thank the authors for their feedback including the parameter sensitivity analysis updated in the PDF - some follow up questions about the mentioned weaknesses and questions that concern me the most below.
>
> 1 - Regarding justification for why progressive extraction will be superior, I understand the actual performance can be dataset dependent, while a mild guarantee should be feasible, e.g., does the Gini coefficient of the residual decrease monotonically across stages under your mask construction, or can you establish a lemma that the expected Gini reduction after two extraction stages is larger than after one?
>
> 2 - Regarding the complexity, even if FFT is the dominant term, operations such as per-channel sorting, GAU’s MLP, and the per-bin sigmoid mask may not be negligible when the number of channels is large. Could you provide a concrete FLOPs breakdown (FFT vs. sorting vs. GAU vs. mask) to justify that these steps remain lower-order in practical multivariate settings?

---

> > ### Author Response · Authors · 2025-11-26
> > **Responses to the follow-up questions**
> >
> > First, thank you for the in-depth review and valuable feedback. Below, we respond to the two questions you raised in order.
> >
> > ## Question 1
> >
> > The proposed lemma does not hold in general. Whether removing high-energy components reduces the Gini coefficient is inherently data dependent, so we cannot provide a distribution-agnostic theoretical guarantee. Moreover, our module does not aim to explicitly minimize the Gini coefficient, nor does it enforce a monotonic reduction of residual Gini across stages. The Gini coefficient is used solely as a descriptive measure of spectral energy imbalance, not as an optimization target.
> >
> > Given this, we rely on a progressive extraction structure for a practical reason rather than a theoretical guarantee. Empirically, some datasets require an optimal number of stages greater than one, and a single-stage design cannot reliably accommodate this variation. When the optimal d is 1, using d = 2 or 3 produces little degradation, indicating that the model is insensitive to setting d above the optimal value. In contrast, when the optimal d is 2 or 3, forcing d = 1 leads to a clear performance drop (Figure 5(b)), creating an asymmetric behavior that favors a progressive design.
> >
> > This asymmetry is consistent with the spectral motivation of our method. Removing the largest energy component narrows the magnitude gap between dominant and weak frequencies, making previously overshadowed low-energy components easier to expose. Applying this process progressively repeatedly reduces the gap without increasing it at later stages. Consequently, using a larger-than-optimal d does not harm performance, whereas using d = 1 on datasets requiring deeper extraction yields insufficient exposure of low-energy components and degrades accuracy.
> >
> > ## Question 2
> >
> > Under the Electricity setting with batch size 32 and horizon 96, the FLOPs of the GiPFE frequency module are as follows:
> > FFT 0.130 GFLOPs, sorting 0.006 GFLOPs, GAU MLP 0.066 GFLOPs, mask 0.013 GFLOPs, totaling 0.214 GFLOPs.
> >
> > With the same input, PatchTST configured with e_layers = 3, n_heads = 4, d_model = 16, d_ff = 128, patch_len = 16, and stride = 8 requires 3.33 GFLOPs.

---

### Official Review · Reviewer_sGQf · 2025-11-01

**Soundness:** 2
**Presentation:** 3
**Contribution:** 2
**Rating:** 4
**Confidence:** 5

**Summary:**

This paper proposes GiPFE, a model-agnostic framework that tackles the problem of spectral energy imbalance in time-series forecasting by progressively extracting dominant frequency components through a Gini-guided strategy. This approach enables a balanced learning process where simple high-energy patterns are captured by lightweight auxiliary heads, while the backbone model focuses on the remaining complex, low-energy structures. The method effectively enhances forecasting generalization across diverse domains and architectures without incurring significant computational overhead.

**Strengths:**

1.The article is clearly written and well-organized, making it highly accessible to readers.
2.Proposes GiPFE for model-agnostic, progressive frequency extraction.
3.Comprehensive experiments demonstrate the model's effectiveness.

**Weaknesses:**

1. In the empirical evaluation of accuracy and efficiency, it is recommended to include a comparative analysis with the one-shot retention of high-energy frequency bands approach. As this method is widely adopted in existing models and GiPFE represents an advancement for the same task, a comprehensive comparison between the two is warranted.
2. In efficiency experiments, the use of a computationally intensive model like PatchTST, which nearly doubles the inference time across most datasets, challenges its characterization as a lightweight approach. It is recommended to expand the efficiency comparison by including other competitive methods, rather than solely benchmarking against the original backbone model.
3. Figure 5(b) demonstrates that increasing the number of extraction layers (d) does not monotonically improve forecasting performance, with noticeable degradation observed in certain configurations. This non-monotonic relationship may stem from potential over-decomposition of frequency components or accumulated prediction errors from multiple auxiliary heads. Furthermore, as the analysis currently relies on a single backbone model and dataset, we recommend expanding the experimental validation to include additional architectures and datasets to better assess the method's generalizability.
4. The paper heuristically assigns high-energy components to lightweight MLP heads and low-energy residuals to the backbone model without empirical or theoretical validation of this design choice. The assignment of MLP heads to high-energy components is justified by the statement they "tend to exhibit simple and clear trends" (Sec. 4.1). However, high-energy components could represent complex periodic patterns or sudden spikes that are not easily captured by simple MLPs. Conversely, some low-energy components might be simple noise.
5. The framework assumes high-energy components are inherently "simple" and low-energy components are "complex," but this may not always hold in practice. There is no analysis that validates this energy-complexity correlation across different datasets.
6. The sequential extraction process might create artificial boundaries in the frequency domain, potentially disrupting important inter-frequency relationships. Each stage isolates components based solely on energy thresholds (Eq. 6-7; Sec. 4.2), but adjacent frequency bins that are separated into different stages might have important phase relationships or joint patterns that are lost in this decomposition. The method doesn't preserve or model cross-component dependencies during prediction.

**Questions:**

see weakness

---

> ### Author Response · Authors · 2025-11-20
> **Response to Reviewer sGQf**
>
> ## Weakness 1
>
> Thank you for the suggestion. We note that our paper already includes a comparison with the one-shot, energy-based selection strategy through FAN, which is, to the best of our knowledge, the only prior method that adopts this design. Most other frequency-domain models rely on predefined high and low frequency splits rather than energy-based selection, leaving FAN as the sole relevant baseline.
>
> Our results in Table 2 compare against FAN directly and show the corresponding performance differences. If you had a specific alternative method in mind, we would be happy to examine it.
>
> ## Weakness 2
>
> We expanded Table 3 by adding PatchTST and iTransformer backbones, as well as SAN as a model-agnostic module. We also report GiPFE with extraction depths d=1,2,3. These updates clarify the inference overhead of GiPFE and enable a fair comparison against both backbone-specific and model-agnostic alternatives.
>
> ## Weakness 3
>
> Our analysis shows that performance generally improves monotonically as d increases, but this trend can break when d becomes excessively large because further decomposition yields little additional benefit once the spectral imbalance has already been reduced. As shown in the revised Figure 5(b), performance remains stable after d≈2–3 and does not exhibit abrupt degradation for larger values, indicating that GiPFE is not overly sensitive to the exact choice of d. To verify that this behavior is consistent, we added experiments using two backbones (DLinear and PatchTST) and two datasets (Weather and Traffic) across multiple horizons, and the same pattern appears in all settings.
>
> ## Weakness 4 & 5
>
> We fully agree that providing additional analysis on this aspect helps readers gain a more comprehensive understanding and strengthens the validity and soundness of our work. To examine the complexity difference between high- and low-energy components, we added a spectral entropy analysis in Appendix C. Table 5 shows that high-energy components consistently have lower entropy across all datasets, while low-energy components have higher entropy.
>
> We also expanded the auxiliary-predictor ablation in Appendix F by comparing our lightweight MLP with a heavier PatchTST-based predictor. As shown in Table 7, the heavier predictor yields only small gains relative to its additional cost, indicating that the high-energy components do not require a heavy model.
>
> ## Weakness 6
>
> We agree that the sequential extraction process may not fully preserve phase relationships or inter-frequency dependencies across adjacent bins.
> In this work, our primary focus is on addressing the fundamental challenge of spectral energy imbalance. We found that mitigating this imbalance yields substantial performance gains across diverse settings, making it a natural first step for the proposed framework. Given this focus, explicitly modeling cross-component dependencies falls beyond the scope of our current study, but it represents a promising next stage in extending our approach.

---

### Author Response · Authors · 2025-12-02
**Supplemental Clarifications for Meta-Review Assessment**

In light of the fact that the reviewer discussion concluded prematurely, we submit the following comment to assist the meta-reviewer in reaching an accurate and well-informed decision.

Our work addresses the problem of spectral energy imbalance in the frequency domain. While prior studies predominantly focus on the imbalance between high- and low-frequency bands, approaches that directly analyze and mitigate energy imbalance across individual frequencies have been largely unexplored. This led to a misconception that our method simply targets high-low imbalance; however, this misunderstanding was clarified through the rebuttal discussion and revisions to the paper.

## Concern 1. Related Work Completeness and Backbone Evaluation
We incorporated the constructive feedback from reviewer gtry by expanding the related work to properly include Fredformer and FilterNet, and we additionally conducted further experiments on these methods. The results show consistent improvements on both backbones, demonstrating that our method preserves the strengths of both backbones while addressing the overlooked issue of frequency-wise energy imbalance. This highlights the value of our method as a model-agnostic enhancement rather than a competing alternative.

## Concern 2. Efficiency and Computational Complexity
Reviewer t2LM raised concerns regarding efficiency and computational complexity. In response, the revision provides detailed experimental settings, additional results, and an explicit complexity analysis, and the lightweight nature of the proposed module was further supported by a FLOPs comparison presented during the discussion.

## Concern 3. Spectral Characteristics Assumption
Reviewer sGQf pointed out that the analysis supporting the assumption that high-energy components are simple while low-energy components are complex was insufficient. The revised paper includes a spectral entropy analysis that empirically confirms this tendency, thereby strengthening the motivation and validity of our proposed approach.

## Concern 4. Progressive Extraction Robustness and Parameter Sensitivity
The choice of progressive extraction over a single-stage design was questioned by reviewer t2LM. Our revised experiments show that it generalizes more reliably across diverse datasets than a single-stage design. We also demonstrate that the depth parameter d is not sensitive: performance remains stable when d is set larger than the optimal value, while fixing d=1 causes substantial degradation on datasets requiring deeper extraction. These results indicate that progressive extraction provides clear benefits where single extraction is insufficient, while maintaining comparable performance when a single stage is optimal. Note that single extraction remains part of our design and contribution.

## Concern 5. Comparison with Standard Measures
Reviewer gtry raised a concern regarding comparisons with standard measures such as spectral normalization and band-energy ratios. The key issue is that these methods are not standard measures for evaluating energy imbalance. Spectral normalization is a GAN-based weight normalization technique unrelated to frequency analysis, and band-energy ratios measure only high-low frequency imbalance rather than frequency-wise imbalance. Comparisons with high-low imbalance baselines were already included in the initial version, representing the closest available alternatives. Moreover, frequency-wise energy imbalance has not been actively studied in prior work, so no established metrics currently exist.

Finally, several weaknesses raised in the reviews originated from misunderstandings of content already addressed in the initial submission. These points were clarified during the discussion.

---

### Meta-Review · Area_Chair_u82R · 2026-01-01

**Summary:**

The proposed GiPFE framework is model-agnostic and it uses of the Gini coefficient to guide progressive frequency extraction. However, the primary concerns include a lack of theoretical proof about why progressive extraction is superior to recent methods, such as recent state-of-the-art frequency modeling papers (e.g., Fredformer, FilterNet). Also, the heuristic assumptions regarding the correlation between spectral energy and signal complexity is not clear. Furthermore, reviewers noted that the extraction process could potentially disrupt important inter-frequency phase relationships, which is also a limitation the authors acknowledged.

**Reviewer Concerns:**

Reviewer t2LM’s comments highlight that the authors failed to provide a theoretical "mild guarantee" (such as monotonic Gini reduction across stages) or a FLOPs breakdown to justify the complexity claims in multivariate settings.
Reviewer gtry’s concerns regarding novelty remain significant; while the authors argue that "magnitude-based energy imbalance" is distinct from "frequency-band imbalance," the conceptual overlap with existing works shows that the paper’s incremental contribution may not meet the bar of acceptance. Additionally, the lack of modeling for cross-component dependencies brings further problem.

**Reviewer Scores:**

I believe they won't change their scores.

---

### Decision · Program_Chairs · 2026-01-26

Reject